# CAUSAL IMITATION LEARNING
# VIA INVERSE REINFORCEMENT LEARNING

**Kangrui Ruan**[*], **Junzhe Zhang**[*], **Xuan Di,** and **Elias Bareinboim**
Columbia University, New York, NY 10027, USA
{kr2910,junzhez,sharon.di,eliasb}@columbia.edu

## ABSTRACT

One of the most common ways children learn when unfamiliar with the environment is by mimicking adults. Imitation learning concerns an imitator learning to behave in an unknown environment from an expert's demonstration; reward signals remain latent to the imitator. This paper studies imitation learning through causal lenses and extends the analysis and tools developed for behavior cloning (Zhang, Kumor, Bareinboim, 2020) to inverse reinforcement learning. First, we propose novel graphical conditions that allow the imitator to learn a policy performing as well as the expert's behavior policy, even when the imitator and the expert's state-action space disagree, and unobserved confounders (UCs) are present. When provided with parametric knowledge about the unknown reward function, such a policy may outperform the expert's. Also, our method is easily extensible and allows one to leverage existing IRL algorithms even when UCs are present, including the multiplicative-weights algorithm (MWAL) (Syed & Schapire, 2008) and the generative adversarial imitation learning (GAIL) (Ho & Ermon, 2016). Finally, we validate our framework by simulations using real-world and synthetic data.

## 1 INTRODUCTION

Reinforcement Learning (RL) has been deployed and shown to perform extremely well in highly complex environments in the past decades (Sutton & Barto, 1998; Mnih et al., 2013; Silver et al., 2016; Berner et al., 2019). One of the critical assumptions behind many of the classical RL algorithms is that the reward signal is fully observed, and the reward function could be well-specified. In many real-world applications, however, it might be impractical to design a suitable reward function that evaluates each and every scenario (Randløv & Alstrøm, 1998; Ng et al., 1999). For example, in the context of human driving, it is challenging to design a precise reward function, and experimenting in the environment could be ill-advised; still, watching expert drivers operating is usually feasible.

In machine learning, the *imitation learning* paradigm investigates the problem of how an agent should behave and learn in an environment with an unknown reward function by observing demonstrations from a human expert (Argall et al., 2009; Billard et al., 2008; Hussein et al., 2017; Osa et al., 2018). There are two major learning modalities that implements IL – *behavioral cloning* (BC) (Widrow, 1964; Pomerleau, 1989; Muller et al., 2006; Mülling et al., 2013; Mahler & Goldberg, 2017) and *inverse reinforcement learning* (IRL) Ng et al. (2000); Ziebart et al. (2008); Ho & Ermon (2016); Fu et al. (2017). BC methods directly mimic the expert's behavior policy by learning a mapping from observed states to the expert's action via supervised learning. Alternatively, IRL methods first learn a potential reward function under which the expert's behavior policy is optimal. The imitator then obtains a policy by employing standard RL methods to maximize the learned reward function. Under some common assumptions, both BC and IRL are able to obtain policies that achieve the expert's performance (Kumor et al., 2021; Swamy et al., 2021). Moreover, when additional parametric knowledge about the reward function is provided, IRL may produce a policy that outperforms the expert's in the underlying environment (Syed & Schapire, 2008; Li et al., 2017; Yu et al., 2020).

For concreteness, consider a learning scenario depicted in Fig. 1a, describing trajectories of human-driven cars collected by drones flying over highways (Krajewski et al., 2018; Etesami & Geiger, 2020). Using such data, we want to learn a policy $X \leftarrow \pi(Z)$ deciding on the acceleration (action) $X \in$

---

[*]Equal contribution.

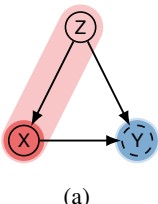 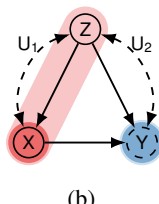 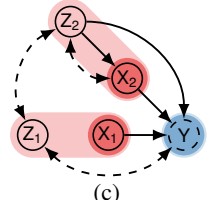 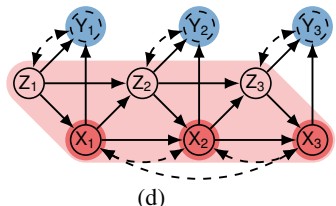

(a)          (b)          (c)          (d)

Figure 1: Causal diagrams where $X$ represents an action (shaded red) and $Y$ represents a latent reward (shaded blue). Input covariates of the policy scope $\mathcal{S}$ are shaded in light red.

$\{0, 1\}$ of the demonstrator car based on velocities and locations $Z$ of surrounding cars. The driving performance is measured by a latent reward signal $Y$. Consider an instance where $Y \leftarrow (1 - X)Z + X(1 - Z)$ and values of $Z$ are drawn uniformly over $\{0, 1\}$. A human expert generates demonstrations following a behavior policy such that $P(X = 1 \mid Z = 0) = 0.6$ and $P(X = 0 \mid Z = 1) = 0.4$. Evaluating the expert's performance gives $\mathbb{E}[Y] = P(X = 1, Z = 0) + P(X = 0, Z = 1) = 0.5$. Now we apply standard IRL algorithms to learn a policy $X \leftarrow \pi(Z)$ so that the imitator's driving performance, denoted by $\mathbb{E}[Y \mid \mathrm{do}(\pi)]$, is at least as good as the expert's performance $\mathbb{E}[Y]$. Detailed derivations of IRL policy are shown in (Ruan et al., 2023, Appendix A). Note that $\mathbb{E}[Y|z, x] = x + z - 2xz$ belongs to a family of reward functions $f_Y(x, z) = \alpha x + \beta z - \gamma xz$, where $0 < \alpha < \gamma$. A typical IRL imitator solves a minimax problem $\min_\pi \max_{f_Y} \mathbb{E}\left[f_Y(X, Z)\right] - \mathbb{E}\left[f_Y(X, Z) \mid \mathrm{do}(\pi)\right]$. The inner step "guesses" a reward function being optimized by the expert; while the outer step learns a policy maximizing the learned reward function. Applying these steps leads to a policy $\pi^* : X \leftarrow \neg Z$ with the expected reward $\mathbb{E}[Y \mid \mathrm{do}(\pi^*)] = 1$, which outperforms the sub-optimal expert.

Despite the performance guarantees provided by existing imitation methods, both BC and IRL rely on the assumption that the expert's input observations match those available to the imitator. More recently, there exists an emerging line of research under the rubric of *causal imitation learning* that augments the imitation paradigm to account for environments consisting of arbitrary causal mechanisms and the aforementioned mismatch between expert and imitator's sensory capabilities (de Haan et al., 2019; Zhang et al., 2020; Etesami & Geiger, 2020; Kumor et al., 2021). Closest to our work, Zhang et al. (2020); Kumor et al. (2021) derived graphical criteria that completely characterize when and how BC could lead to successful imitation even when the agents perceive reality differently. Still, it is unclear how to perform IRL-type training if some expert's observed states remain latent to the imitator, which leads to the presence of unobserved confounding (UCs) in expert's demonstrations. Perhaps surprisingly, naively applying IRL methods when UCs are present does not necessarily lead to satisfactory performance, even when the expert itself behaves optimally.

To witness, we now modify the previous highway driving scenario to demonstrate the challenges of UCs. In reality, covariates $Z$ (i.e., velocities and location) are also affected by the car horn $U_1$ of surrounding vehicles and the wind condition $U_2$. However, due to the different perspectives of drones (recording from the top), such critical information (i.e, $U_1, U_2$ ) is not recorded by the camera and thus remains unobserved. Fig. 1b graphically describes this modified learning setting. More specifically, consider an instance where $Z \leftarrow U_1 \oplus U_2$, $Y \leftarrow \neg X \oplus Z \oplus U_2$; $\oplus$ is the *exclusive-or* operator; and values of $U_1$ and $U_2$ are drawn uniformly over $\{0, 1\}$. An expert driver, being able to hear the car horn $U_1$, follows a behavior policy $X \leftarrow U_1$ and achieves the optimal performance $\mathbb{E}[Y] = 1$. Meanwhile, observe that $\mathbb{E}[Y|z, x] = 1$ belongs to a family of reward functions $f_Y(x, z) = \alpha$ (where $\alpha > 0$). Solving $\min_\pi \max_{f_Y} \mathbb{E}\left[f_Y(X, Z)\right] - \mathbb{E}\left[f_Y(X, Z) \mid \mathrm{do}(\pi)\right]$ leads to an IRL policy $\pi^*$ with expected reward $\mathbb{E}[Y|\mathrm{do}(\pi^*)] = 0.5$, which is far from the expert's optimal performance $\mathbb{E}[Y] = 1$.

After all, a question that naturally arises is, under what conditions an IRL imitator procedure can perform well when UCs are present, and there is a mismatch between the perception of the two agents? In this paper, we answer this question and, more broadly, investigate the challenge of performing IRL through causal lenses. In particular, our contributions are summarized as follows. (1) We provide a novel, causal formulation of the inverse reinforcement learning problem. This formulation allows one to formally study and understand the conditions under which an IRL policy is learnable, including in settings where UCs cannot be ruled out a priori. (2) We derive a new graphical condition for deciding whether an imitating policy can be computed from the available data and knowledge, which provides a robust generalization of current IRL algorithms to non-Markovian settings, including GAIL (Ho & Ermon, 2016) and MWAL (Syed & Schapire, 2008). (3) Finally, we move beyond this graphical condition and develop an effective IRL algorithm for structural causal models (Pearl, 2000) with

arbitrary causal relationships. Due to the space constraints, all proofs are provided in (Ruan et al., 2023, Appendix B). For a more detailed survey on imitation learning and causal inference, we refer readers to (Ruan et al., 2023, Appendix E).

## 1.1 Preliminaries

We use capital letters to denote random variables ($X$) and small letters for their values ($x$). $\mathscr{D}_X$ represents the domain of $X$ and $\mathscr{P}_X$ the space of probability distributions over $\mathscr{D}_X$. For a set $\boldsymbol{X}$, let $|\boldsymbol{X}|$ denote its dimension. The probability distribution over variables $\boldsymbol{X}$ is denoted by $P(\boldsymbol{X})$. Similarly, $P(\boldsymbol{Y} \mid \boldsymbol{X})$ represents a set of conditional distributions $P(\boldsymbol{Y} \mid \boldsymbol{X} = \boldsymbol{x})$ for all realizations $\boldsymbol{x}$. We use abbreviations $P(\boldsymbol{x})$ for probabilities $P(\boldsymbol{X} = \boldsymbol{x})$; so does $P(\boldsymbol{Y} = \boldsymbol{y} \mid \boldsymbol{X} = \boldsymbol{x}) = P(\boldsymbol{y} \mid \boldsymbol{x})$. Finally, indicator function $\mathbb{1}\{\boldsymbol{Z} = \boldsymbol{z}\}$ returns 1 if $\boldsymbol{Z} = \boldsymbol{z}$ holds true; otherwise 0.

The basic semantic framework of our analysis rests on *structural causal models* (SCMs) (Pearl, 2000, Ch. 7). An SCM $M$ is a tuple $\langle \boldsymbol{U}, \boldsymbol{V}, \mathcal{F}, P(\boldsymbol{U}) \rangle$ with $\boldsymbol{V}$ the set of endogenous, and $\boldsymbol{U}$ exogenous variables. $\mathcal{F}$ is a set of structural functions s.t. for $f_V \in \mathcal{F}$, $V \leftarrow f_V(\boldsymbol{pa}_V, \boldsymbol{u}_V)$, with $\boldsymbol{PA}_V \subseteq \boldsymbol{V}, \boldsymbol{U}_V \subseteq \boldsymbol{U}$. Values of $\boldsymbol{U}$ are drawn from an exogenous distribution $P(\boldsymbol{U})$, inducing distribution $P(\boldsymbol{V})$ over endogenous variables $\boldsymbol{V}$. Since the learner can observe only a subset of endogenous variables, we split $\boldsymbol{V}$ into a partition $\boldsymbol{O} \cup \boldsymbol{L}$ where variable $\boldsymbol{O} \subseteq \boldsymbol{V}$ are observed and $\boldsymbol{L} = \boldsymbol{V} \setminus \boldsymbol{O}$ remain latent to the leaner. The marginal distribution $P(\boldsymbol{O})$ is thus referred to as the *observational distribution*. An *atomic intervention* on a subset $\boldsymbol{X} \subseteq \boldsymbol{V}$, denoted by do($\boldsymbol{x}$), is an operation where values of $\boldsymbol{X}$ are set to constants $\boldsymbol{x}$, replacing the functions $f_{\boldsymbol{X}} = \{f_X : \forall X \in \boldsymbol{X}\}$ that would normally determine their values. For an SCM $M$, let $M_{\boldsymbol{x}}$ be a submodel of $M$ induced by intervention do($\boldsymbol{x}$). For a set $\boldsymbol{Y} \subseteq \boldsymbol{V}$, the interventional distribution $P(\boldsymbol{s}|\text{do}(\boldsymbol{x}))$ induced by do($\boldsymbol{x}$) is defined as the distribution over $\boldsymbol{Y}$ in the submodel $M_{\boldsymbol{x}}$, i.e., $P_M(\boldsymbol{Y}|\text{do}(\boldsymbol{x})) \triangleq P_{M_{\boldsymbol{x}}}(\boldsymbol{Y})$. We leave $M$ implicit when it is obvious from the context.

Each SCM $M$ is associated with a causal diagram $\mathcal{G}$ which is a directed acyclic graph where (e.g., see Fig. 1) solid nodes represent observed variables $\boldsymbol{O}$, dashed nodes represent latent variables $\boldsymbol{L}$, and arrows represent the arguments $\boldsymbol{PA}_V$ of each function $f_V \in \mathcal{F}$. Exogenous variables $\boldsymbol{U}$ are not explicitly shown; a bi-directed arrow between nodes $V_i$ and $V_j$ indicates the presence of an unobserved confounder (UC) affecting both $V_i$ and $V_j$. We will use family abbreviations to represent graphical relationships such as parents, children, descendants, and ancestors. For example, the set of parent nodes of $\boldsymbol{X}$ in $\mathcal{G}$ is denoted by $pa(\boldsymbol{X})_{\mathcal{G}} = \cup_{X \in \boldsymbol{X}} pa(X)_{\mathcal{G}}$; $ch$, $de$ and $an$ are similarly defined. Capitalized versions $Pa, Ch, De, An$ include the argument as well, e.g. $Pa(\boldsymbol{X})_{\mathcal{G}} = pa(\boldsymbol{X})_{\mathcal{G}} \cup \boldsymbol{X}$. For a subset $\boldsymbol{X} \subseteq \boldsymbol{V}$, the subgraph obtained from $\mathcal{G}$ with edges outgoing from $\boldsymbol{X}$ / incoming into $\boldsymbol{X}$ removed is written as $\mathcal{G}_{\underline{\boldsymbol{X}}}/\mathcal{G}_{\overline{\boldsymbol{X}}}$ respectively. $\mathcal{G}_{[\boldsymbol{X}]}$ is a subgraph of $\mathcal{G}$ containing only nodes $\boldsymbol{X}$ and edges among them. A path from a node $X$ to a node $Y$ in $\mathcal{G}$ is a sequence of edges, which does not include a particular node more than once. Two sets of nodes $\boldsymbol{X}, \boldsymbol{Y}$ are said to be d-separated by a third set $\boldsymbol{Z}$ in a DAG $\mathcal{G}$, denoted by $(\boldsymbol{X} \perp\!\!\!\perp \boldsymbol{Y}|\boldsymbol{Z})_{\mathcal{G}}$, if every edge path from nodes in $\boldsymbol{X}$ to nodes in $\boldsymbol{Y}$ is "blocked" by nodes in $\boldsymbol{Z}$. The criterion of blockage follows (Pearl, 2000, Def. 1.2.3). For a more detailed survey on SCMs, we refer readers to (Pearl, 2000; Bareinboim et al., 2022).

## 2 Causal Inverse Reinforcement Learning

We investigate the sequential decision-making setting concerning a set of actions $\boldsymbol{X}$, a series of covariates $\boldsymbol{Z}$, and a latent reward $Y$ in an SCM $M$. An expert (e.g., a physician, driver), operating in SCM $M$, selects actions following a *behavior policy*, which is the collection of structural functions $f_{\boldsymbol{X}} = \{f_X \mid X \in \boldsymbol{X}\}$. The expert's performance is evaluated as the expected reward $\mathbb{E}[Y]$. On the other hand, a learning agent (i.e., the imitator) intervenes on actions $\boldsymbol{X}$ following an ordering $X_1 \prec \cdots \prec X_n$; each action $X_i$ is associated with a set of features $\boldsymbol{PA}_i^* \subseteq \boldsymbol{O} \setminus \{X_i\}$. A policy $\boldsymbol{\pi}$ over actions $\boldsymbol{X}$ is a sequence of decision rules $\boldsymbol{\pi} = \{\pi_1, \ldots, \pi_n\}$. Each decision rule $\pi_i(X_i \mid \boldsymbol{Z}_i)$ is a probability distribution over an action $X_i \in \boldsymbol{X}$, conditioning on values of a set of covariates $\boldsymbol{Z}_i \subseteq \boldsymbol{PA}_i^*$. Such policies $\boldsymbol{\pi}$ are also referred to as dynamic treatment regimes (Murphy et al., 2001; Chakraborty & Murphy, 2014), which generalize personalized medicine to time-varying treatment settings in healthcare, in which treatment is repeatedly tailored to a patient's dynamic state.

A *policy intervention* on actions $\boldsymbol{X}$ following a policy $\boldsymbol{\pi}$, denoted by do($\boldsymbol{\pi}$), entails a submodel $M_{\boldsymbol{\pi}}$ from a SCM $M$ where structural functions $f_{\boldsymbol{X}}$ associated with $\boldsymbol{X}$ (i.e., the expert's behavior policy) are replaced with decision rules $X_i \sim \pi_i(X_i \mid \boldsymbol{Z}_i)$ for every $X_i \in \boldsymbol{X}$. A critical assumption

throughout this paper is that submodel $M_{\boldsymbol{\pi}}$ does not contain any cycles. Similarly, the interventional distribution $P(\boldsymbol{V} \mid \mathrm{do}(\boldsymbol{\pi}))$ induced by policy $\boldsymbol{\pi}$ is defined as the joint distribution over $\boldsymbol{V}$ in $M_{\boldsymbol{\pi}}$.

Throughout this paper, detailed parametrizations of the underlying SCM $M$ are assumed to be *unknown* to the agent. Instead, the agent has access to the **input**: (1) a causal diagram $\mathcal{G}$ associated with $M$, and (2) the expert's demonstrations, summarized as the observational distribution $P(\boldsymbol{O})$. The goal of the agent is to **output** an imitating policy $\boldsymbol{\pi}^*$ that achieves the expert's performance.

**Definition 1.** For an SCM $M = \langle \boldsymbol{U}, \boldsymbol{V}, \mathcal{F}, P(\boldsymbol{U}) \rangle$, an *imitating policy* $\boldsymbol{\pi}^*$ is a policy such that its expected reward is lower bounded by the expert's reward, i.e., $\mathbb{E}_M[Y \mid \mathrm{do}(\boldsymbol{\pi}^*)] \geq \mathbb{E}_M[Y]$.

In words, the right-hand side is the expert's performance that the agent wants to achieve, while the left-hand side is the real reward experienced by the agent. The challenge in imitation learning arises from the fact that the reward $Y$ is not specified and latent, i.e., $Y \notin \boldsymbol{O}$. This precludes approaches that identify $\mathbb{E}[Y|\mathrm{do}(\boldsymbol{\pi})]$ directly from the demonstration data (e.g., through the do- or soft-do-calculus Pearl (2000); Correa & Bareinboim (2020)).

There exist methods in the literature for finding an imitating policy in Def. 1. Before describing their details, we first introduce some necessary concepts. For any policy $\boldsymbol{\pi}$, we summarize its associated state-action domain using a sequence of pairs of variables called a policy scope $\mathcal{S}$.

**Definition 2** (Lee & Bareinboim (2020)). For an SCM $M$, a policy scope $\mathcal{S}$ (for short, scope) over actions $\boldsymbol{X}$ is a sequence of tuples $\{\langle X_i, \boldsymbol{Z}_i \rangle\}_{i=1}^n$ where $\boldsymbol{Z}_i \subseteq \boldsymbol{PA}_i^*$ for every $X_i \in \boldsymbol{X}$.

We will consistently use $\boldsymbol{\pi} \sim \mathcal{S}$ to denote a policy $\boldsymbol{\pi}$ associated with scope $\mathcal{S}$. For example, consider a policy scope $\mathcal{S} = \{\langle X_1, \{Z_1\} \rangle, \langle X_2, \{Z_2\} \rangle\}$ over actions $X_1, X_2$ in Fig. 1c. A policy $\boldsymbol{\pi} \sim \mathcal{S}$ is a sequence of distributions $\boldsymbol{\pi} = \{\pi_1(X_1 \mid Z_1), \pi_2(X_2 \mid Z_2)\}$.

Zhang et al. (2020); Kumor et al. (2021) provide a graphical condition that is sufficient for learning an imitating policy via behavioral cloning (BC) provided with a causal diagram $\mathcal{G}$. For a policy scope $\mathcal{S} = \{\langle X_i, \boldsymbol{Z}_i \rangle\}_{i=1}^n$, let $\mathcal{G}^{(i)}$, $i = 1, \ldots, n$, denote a manipulated graph obtained from $\mathcal{G}$ by the following steps: for all $j = i+1, \ldots, n$, (1) remove arrows coming into every action $X_j$; and (2) add direct arrows from nodes in $\boldsymbol{Z}_j$ to $X_j$. Formally, the *sequential $\pi$-backdoor* criterion is defined as:

**Definition 3** (Kumor et al. (2021)). Given a causal diagram $\mathcal{G}$, a policy scope $\mathcal{S} = \{\langle X_i, \boldsymbol{Z}_i \rangle\}_{i=1}^n$ is said to satisfy the *sequential $\pi$-backdoor* criterion in $\mathcal{G}$ (for short, $\pi$-backdoor admissible) if at each $X_i \in \boldsymbol{X}$, one of the following conditions hold: (1) $X_i$ is not an ancestor of $Y$ in $\mathcal{G}^{(i)}$, i.e., $X \notin An(Y)_{\mathcal{G}^{(i)}}$; or (2) $\boldsymbol{Z}_i$ blocks all backdoor path from $X_i$ to $Y$ in $\mathcal{G}^{(i)}$, i.e., $(Y \perp\!\!\!\perp X_i | \boldsymbol{Z}_i)$ in $\mathcal{G}^{(i)}_{\underline{X_i}}$.

(Kumor et al., 2021) showed that whenever a $\pi$-backdoor admissible scope $\mathcal{S}$ is available, one could learn an imitating policy $\boldsymbol{\pi}^* \sim \mathcal{S}$ by setting $\pi_i^*(x_i \mid \boldsymbol{z}_i) = P(x_i \mid \boldsymbol{z}_i)$ for every action $X_i \in \boldsymbol{X}$. For instance, consider the causal diagram $\mathcal{G}$ in Fig. 1c. Scope $\mathcal{S} = \{\langle X_1, \{Z_1\} \rangle, \langle X_2, \{Z_2\} \rangle\}$ is $\pi$-backdoor admissible since $(X_1 \perp\!\!\!\perp Y | Z_1)$ and $(X_2 \perp\!\!\!\perp Y | Z_2)$ hold in $\mathcal{G}$, which is a super graph containing both manipulated $\mathcal{G}^{(1)}$ and $\mathcal{G}^{(2)}$. An imitating policy $\boldsymbol{\pi}^* = \{\pi_1^*, \pi_2^*\}$ is thus obtainable by setting $\pi_1^*(X_1 \mid Z_1) = P(X_1 \mid Z_1)$ and $\pi_2^*(X_2 \mid Z_2) = P(X_2 \mid Z_2)$. While impressive, a caveat of their results is that the performance of the imitator is restricted by that of the expert, i.e., $\mathbb{E}[Y \mid \mathrm{do}(\boldsymbol{\pi}^*)] = \mathbb{E}[Y]$. In other words, causal BC provides an efficient way to mimic the expert's performance. If the expert's behavior is far from optimal, the same will hold for the learning agent.

## 2.1 MINIMAL SEQUENTIAL BACKDOOR CRITERION

To circumvent this issue, we take a somewhat different approach to causal imitation by incorporating the principle of inverse reinforcement learning (IRL) principle. Following the game-theoretic approach (Syed & Schapire, 2008), we formulate the problem as learning to play a two-player zero-sum game in which the agent chooses a policy, and the nature chooses an SCM instance. A key property of this algorithm is that it allows us to incorporate prior parametric knowledge about the latent reward signal. When such knowledge is informative, our algorithm is about to obtain a policy that could significantly outperform the expert with respect to the unknown causal environment, while at the same time are guaranteed to be no worse. Formally, let $\mathscr{M} = \{\forall M \mid \mathcal{G}_M = \mathcal{G}, P_M(\boldsymbol{O}) = P(\boldsymbol{O})\}$ denote the set of SCMs compatible with both the causal diagram $\mathcal{G}$ and the observational distribution $P(\boldsymbol{O})$. Fix a policy scope $\mathcal{S}$. Now consider the optimization problem defined as follows.

$$\nu^* = \min_{\boldsymbol{\pi} \sim \mathcal{S}} \max_{M \in \mathscr{M}} \mathbb{E}_M[Y] - \mathbb{E}_M[Y \mid \mathrm{do}(\boldsymbol{\pi})]. \tag{1}$$

The inner maximization in the above equation can be viewed as an *causal IRL* step where we attempt to "guess" a worst-case SCM $\hat{M}$ compatible with $\mathcal{G}$ and $P(\boldsymbol{O})$ that prioritizes the expert's policy. That is, the gap in the performance between the expert's and the imitator's policies is maximized. Meanwhile, since the expert's reward $\mathbb{E}_M[Y]$ is not affected by the imitator's policy $\boldsymbol{\pi}$, the outer minimization is equivalent to a planning step that finds a policy $\boldsymbol{\pi}^*$ optimizing the learned SCM $\hat{M}$. Obviously, the solution $\boldsymbol{\pi}^*$ is an imitating policy if gap $\nu^* = 0$. In cases where the expert is sub-optimal, i.e., $E_{\hat{M}}[Y] < E_{\hat{M}}[Y \mid do(\boldsymbol{\pi})]$ for some policies $\boldsymbol{\pi}$, we may have $\nu^* < 0$. That is, the policy $\boldsymbol{\pi}^*$ will dominate the expert's policy $f_{\boldsymbol{X}}$ regardless of parametrizations of SCM $M$ in the worst-case scenario. In other words, $\boldsymbol{\pi}^*$ to some extent ignores the sub-optimal expert, and instead exploits prior knowledge about the underlying model.

Despite the clear semantics in terms of causal models, the optimization problem in Eq. (1) requires the learner to search over all possible SCMs compatible with the causal diagram $\mathcal{G}$ and observational distribution $P(\boldsymbol{O})$. In principle, it entails a quite challenging search since one does not have access to the parametric forms of the underlying structural functions $\mathcal{F}$ nor the exogenous distribution $P(\boldsymbol{U})$. It is not clear how the existing optimization procedures can be used.

In this paper, we will develop novel methods to circumvent this issue, thus leading to effective imitating policies. Our first algorithm relies on a refinement of the sequential $\pi$-backdoor, based on the concept of minimality. A subscope $\mathcal{S}'$ of a policy scope $\mathcal{S} = \{\langle X_i, \boldsymbol{Z}_i \rangle\}_{i=1}^n$, denoted by $\mathcal{S}' \subseteq \mathcal{S}$, is a sequence $\{\langle X_i, \boldsymbol{Z}_i' \rangle\}_{i=1}^n$ where $\boldsymbol{Z}_i' \subseteq \boldsymbol{Z}_i$ for every $X_i \in \boldsymbol{X}$. A proper subscope $\mathcal{S}' \subset \mathcal{S}$ is a subscope in $\mathcal{S}$ other than $\mathcal{S}$ itself. The minimal $\pi$-backdoor admissible scope is defined as follows.

**Definition 4.** Given a causal diagram $\mathcal{G}$, a $\pi$-backdoor admissible scope $\mathcal{S}$ is said to be *minimal* if there exists no proper subscope $\mathcal{S}' \subset \mathcal{S}$ satisfying the sequential $\pi$-backdoor in $\mathcal{G}$.

**Theorem 1.** *Given a causal diagram $\mathcal{G}$, if there exists a minimal $\pi$-backdoor admissible scope $\mathcal{S} = \{\langle X_i, \boldsymbol{Z}_i \rangle\}_{i=1}^n$ in $\mathcal{G}$, consider the following conditions:*

1. *Let effective actions $\boldsymbol{X}^* = \boldsymbol{X} \cap An(Y)_{\mathcal{G}_{\mathcal{S}}}$ and effective covariates $\boldsymbol{Z}^* = \bigcup_{X_i \in \boldsymbol{X}^*} \boldsymbol{Z}_i$;*
2. *For $i = 1, \ldots, n+1$, let $\boldsymbol{X}_{<i}^* = \{\forall X_j \in \boldsymbol{X}^* \mid j < i\}$ and $\boldsymbol{Z}_{<i}^* = \bigcup_{X_j \in \boldsymbol{X}_{<i}^*} \boldsymbol{Z}_j$.*

*Then, for any policy $\boldsymbol{\pi} \sim \mathcal{S}$, the expected reward $\mathbb{E}[Y \mid do(\boldsymbol{\pi})]$ is computable from $P(\boldsymbol{O}, Y)$ as:*

$$\mathbb{E}[Y \mid do(\boldsymbol{\pi})] = \sum_{\boldsymbol{x}^*, \boldsymbol{z}^*} \mathbb{E}[Y \mid \boldsymbol{x}^*, \boldsymbol{z}^*] \rho_{\boldsymbol{\pi}}(\boldsymbol{x}^*, \boldsymbol{z}^*) \tag{2}$$

*where the* occupancy measure $\rho_{\boldsymbol{\pi}}(\boldsymbol{x}^*, \boldsymbol{z}^*) = \prod_{X_i \in \boldsymbol{X}^*} P\left(\boldsymbol{z}_i \mid \boldsymbol{x}_{<i}^*, \boldsymbol{z}_{<i}^*\right) \pi_i(x_i \mid \boldsymbol{z}_i)$.

To illustrate, consider again the causal diagram $\mathcal{G}$ in Fig. 1c; the manipulated diagram $\mathcal{G}^{(2)} = \mathcal{G}$ and $\mathcal{G}^{(1)}$ is obtained from $\mathcal{G}$ by removing $Z_2 \leftrightarrow X_2$. While scope $\mathcal{S}_1 = \{\langle X_1, \{Z_1\} \rangle, \langle X_2, \{Z_2\} \rangle\}$ satisfies the sequential $\pi$-backdoor, it is not minimal since $(X_1 \perp\!\!\!\perp Y)$ in $\mathcal{G}_{\underline{X_1}}^{(1)}$. On the other hand, $\mathcal{S}_2 = \{\langle X_1, \emptyset \rangle, \langle X_2, \{Z_2\} \rangle\}$ is minimal $\pi$-backdoor admissible since $(X_2 \perp\!\!\!\perp Y \mid Z_2)$ holds true in $\mathcal{G}_{\underline{X_2}}^{(2)}$; and the covariate set $\{Z_2\}$ is minimal due to the presence of the backdoor path $X_2 \leftarrow Z_2 \rightarrow Y$.

Let us focus on the minimal $\pi$-backdoor admissible scope $\mathcal{S}_2$. Note that $\mathcal{G}_{\mathcal{S}_2}$ is a subgraph obtained from $\mathcal{G}$ by removing the bi-directed arrow $Z_2 \leftrightarrow X_2$. We must have effective actions $\boldsymbol{X}^* = \{X_1, X_2\}$ and effective covariates $\boldsymbol{Z}^* = \{Z_2\}$. Therefore, $\boldsymbol{Z}_{<1}^* = \boldsymbol{Z}_{<2}^* = \emptyset$ and $\boldsymbol{Z}_{<3}^* = \{Z_2\}$. For any policy $\boldsymbol{\pi} \sim \mathcal{S}_2$, Thm. 1 implies $\mathbb{E}[Y \mid do(\boldsymbol{\pi})] = \sum_{x_1, x_2, z_2} \mathbb{E}[Y \mid x_1, x_2, z_2] P(z_2|x_1) \pi_2(x_2|z_2) \pi(x_1)$. On the other hand, the same result in Thm. 1 does not necessarily hold for a non-minimal $\pi$-backdoor admissible scope. For instance, consider again the non-minimal scope $\mathcal{S}_1 = \{\langle X_1, \{Z_1\} \rangle, \langle X_2, \{Z_2\} \rangle\}$. The expected reward $\mathbb{E}[Y \mid do(\boldsymbol{\pi})]$ of a policy $\boldsymbol{\pi} \sim \mathcal{S}_2$ is not computable from Eq. (2), and is ultimately not identifiable from distribution $P(\boldsymbol{O}, Y)$ in $\mathcal{G}$ (Tian, 2008).

## 2.2 IMITATION VIA INVERSE REINFORCEMENT LEARNING

Once a minimal $\pi$-backdoor admissible scope $\mathcal{S}$ is found, there exist effective procedures to solve for an imitating policy in Eq. (1). Let $\mathscr{R}$ be a hypothesis class containing all expected rewards $\mathbb{E}_M[Y \mid \boldsymbol{x}^*, \boldsymbol{z}^*]$ compatible with candidate SCMs $M \in \mathscr{M}$, i.e., $\mathscr{R} = \{\mathbb{E}_M[Y \mid \boldsymbol{x}^*, \boldsymbol{z}^*] \mid \forall M \in \mathscr{M}\}$. Applying the identification formula in Thm. 1 reduces the optimization problem in Eq. (1) as follows:

$$\nu^* = \min_{\boldsymbol{\pi} \sim \mathcal{S}} \max_{r \in \mathscr{R}} \sum_{\boldsymbol{x}^*, \boldsymbol{z}^*} r(\boldsymbol{x}^*, \boldsymbol{z}^*) \left(\rho(\boldsymbol{x}^*, \boldsymbol{z}^*) - \rho_{\boldsymbol{\pi}}(\boldsymbol{x}^*, \boldsymbol{z}^*)\right) \tag{3}$$

where the expert's occupancy measure $\rho(\boldsymbol{x}^*, \boldsymbol{z}^*) = P(\boldsymbol{x}^*, \boldsymbol{z}^*)$ and the agent's occupancy measure $\rho_{\boldsymbol{\pi}}(\boldsymbol{x}^*, \boldsymbol{z}^*)$ is given by Eq. (2). The above minimax problem is solvable using standard IRL algorithms. The identification result in Thm. 1 ensures that the learned policy applies to any SCM compatible with the causal diagram and the observational data, thus robust to the unobserved confounding bias in the expert's demonstrations. Henceforth, we will consistently refer to Eq. (3) as the ***canonical equation of causal IRL***. In this paper, we solve for an imitating policy $\boldsymbol{\pi}^*$ in Eq. (3) using state-of-the-art IRL algorithms, provided with common choices of parametric reward functions. These algorithms include the multiplicative-weights algorithm (MWAL) (Syed & Schapire, 2008) and the generative adversarial imitation learning (GAIL) (Ho & Ermon, 2016). We refer readers to Algs. 3 and 4 in (Ruan et al., 2023, Appendix C) for more discussions on the pseudo-code and implementation details.

**Causal MWAL**   (Abbeel & Ng, 2004; Syed & Schapire, 2008) study IRL in Markov decision processes where the reward function $r(\boldsymbol{x}^*, \boldsymbol{z}^*)$ is a linear combination of $k$-length *feature expectations* vectors $\boldsymbol{\phi}(\boldsymbol{x}^*, \boldsymbol{z}^*)$. Particularly, let $r(\boldsymbol{x}^*, \boldsymbol{z}^*) = \boldsymbol{w} \cdot \boldsymbol{\phi}(\boldsymbol{x}^*, \boldsymbol{z}^*)$ for a coefficient vector $\boldsymbol{w}$ contained in a convex set $\mathbb{S}^k = \left\{ \boldsymbol{w} \in \mathbb{R}^k \mid \|\boldsymbol{w}\|_1 = 1 \text{ and } \boldsymbol{w} \succeq \boldsymbol{0} \right\}$. Let $\boldsymbol{\phi}^{(i)}$ be the $i$-th component of feature vector $\boldsymbol{\phi}$ and let deterministic policies with scope $\mathcal{S}$ be ordered by $\boldsymbol{\pi}^{(1)}, \dots, \boldsymbol{\pi}^{(n)}$. The canonical equation in Eq. (3) is reducible to a two-person zero-sum matrix game under linearity.

**Proposition 1.** *For a hypothesis class $\mathscr{R} = \{r = \boldsymbol{w} \cdot \boldsymbol{\phi} \mid \boldsymbol{w} \in \mathbb{S}^k\}$, the solution $\nu^*$ of the canonical equation in Eq. (3) is obtainable by solving the following minimax problem:*

$$\nu^* = \min_{\boldsymbol{\pi} \sim \mathcal{S}} \max_{\boldsymbol{w} \in \mathbb{S}^k} \boldsymbol{w}^\top \boldsymbol{G} \boldsymbol{\pi}, \tag{4}$$

*where $\boldsymbol{G}$ is a $k \times n$ matrix given by $\boldsymbol{G}(i,j) = \sum_{\boldsymbol{x}^*, \boldsymbol{z}^*} \boldsymbol{\phi}^{(i)}(\boldsymbol{x}^*, \boldsymbol{z}^*) \left( \rho(\boldsymbol{x}^*, \boldsymbol{z}^*) - \rho_{\boldsymbol{\pi}^{(j)}}(\boldsymbol{x}^*, \boldsymbol{z}^*) \right).$*

There exist effective multiplicative weights algorithms for solving the matrix game in Eq. (4), including MW (Freund & Schapire, 1999) and MWAL (Syed & Schapire, 2008).

**Causal GAIL**   (Ho & Ermon, 2016) introduces the GAIL algorithm for learning an imitating policy in Markov decision processes with a general family of non-linear reward functions. In particular, $r(\boldsymbol{x}^*, \boldsymbol{z}^*)$ takes values in the real space $\mathbb{R}$, i.e., $r \in \mathbb{R}^{\boldsymbol{X}^*, \boldsymbol{Z}^*}$ where $\mathbb{R}^{\boldsymbol{X}^*, \boldsymbol{Z}^*} = \{r : \mathscr{D}_{\boldsymbol{X}^*} \times \mathscr{D}_{\boldsymbol{Z}^*} \mapsto \mathbb{R}\}$. The complexity of reward function $r$ is penalized by a convex regularization function $\psi(r)$, i.e.,

$$\nu^* = \min_{\boldsymbol{\pi} \sim \mathcal{S}} \max_{r \in \mathbb{R}^{\boldsymbol{X} \times \boldsymbol{Z}}} \sum_{\boldsymbol{x}^*, \boldsymbol{z}^*} r(\boldsymbol{x}^*, \boldsymbol{z}^*) \left( \rho(\boldsymbol{x}^*, \boldsymbol{z}^*) - \rho_{\boldsymbol{\pi}}(\boldsymbol{x}^*, \boldsymbol{z}^*) \right) - \psi(r) \tag{5}$$

Henceforth, we will consistently refer to Eq. (5) as the *penalized canonical equation* of causal IRL. It is often preferable to solve its conjugate form. Formally,

**Proposition 2.** *For a hypothesis class $\mathscr{R} = \{r : \mathscr{D}_{\boldsymbol{X}^*} \times \mathscr{D}_{\boldsymbol{Z}^*} \mapsto \mathbb{R}\}$ regularized by $\psi$, the solution $\nu^*$ of the penalized canonical equation in Eq. (5) is obtainable by solving the following problem:*

$$\nu^* = \min_{\boldsymbol{\pi} \sim \mathcal{S}} \psi^* \left( \rho - \rho_{\boldsymbol{\pi}} \right) \tag{6}$$

*where $\psi^*$ be a conjugate function of $\psi$ and is given by $\psi^* = \max_{r \in \mathbb{R}^{\boldsymbol{X} \times \boldsymbol{Z}}} a^\top r - \psi(r)$.*

Eq. (6) seeks a policy $\boldsymbol{\pi}$ which minimizes the divergence of the occupancy measures between the imitator and the expert, as measured by the function $\psi^*$. The computational framework of generative adversarial networks (Goodfellow et al., 2014) provides an effective approach to solve such a matching problem, e.g., the GAIL algorithm (Ho & Ermon, 2016).

## 3   CAUSAL IMITATION WITHOUT SEQUENTIAL BACKDOOR

In this section, we investigate causal IRL beyond the condition of minimal sequential $\pi$-backdoor. Observe that the key to the reduction of the canonical causal IRL equation in Eq. (3) lies in the identification of expected rewards $\mathbb{E}[Y \mid \mathrm{do}(\boldsymbol{\pi})]$ had the latent reward $Y$ been observed. Next we will study general conditions under which $\mathbb{E}[Y \mid \mathrm{do}(\boldsymbol{\pi})]$ is uniquely discernible from distribution $P(\boldsymbol{O}, Y)$ in the causal diagram $\mathcal{G}$, called the *identifiability* of causal effects (Pearl, 2000, Def. 3.2.4).

**Definition 5** (Identifiability). Given a causal diagram $\mathcal{G}$ and a policy $\boldsymbol{\pi} \sim \mathcal{S}$, the expected reward $\mathbb{E}[Y \mid \mathrm{do}(\boldsymbol{\pi})]$ is said to be identifiable from distribution $P(\boldsymbol{O}, Y)$ in $\mathcal{G}$ if $\mathbb{E}[Y \mid \mathrm{do}(\boldsymbol{\pi})]$ is uniquely computable from $P(\boldsymbol{O}, Y)$ in any SCM $M$ compatible with $\mathcal{G}$.

We say a policy scope $\mathcal{S}$ is identifiable (from $P(\boldsymbol{O}, Y)$ in $\mathcal{G}$) if for all policies $\boldsymbol{\pi} \sim \mathcal{S}$, the corresponding expected rewards $\mathbb{E}[Y \mid do(\boldsymbol{\pi})]$ are identifiable from $P(\boldsymbol{O}, Y)$ in $\mathcal{G}$. Our next result shows that whenever an identifiable policy scope $\mathcal{S}$ is found, one could always reduce the causal IRL problem to the canonical optimization equation in Eq. (3).

**Theorem 2.** *Given a causal diagram $\mathcal{G}$, a policy scope $\mathcal{S}$ is identifiable from $P(\boldsymbol{O}, Y)$ in $\mathcal{G}$ if and only if for any policy $\boldsymbol{\pi} \sim \mathcal{S}$, the expected reward $\mathbb{E}[Y \mid do(\boldsymbol{\pi})]$ is computable from $P(\boldsymbol{O}, Y)$ as*

$$\mathbb{E}[Y \mid do(\boldsymbol{\pi})] = \sum_{\boldsymbol{x}^*, \boldsymbol{z}^*} \mathbb{E}[Y \mid \boldsymbol{x}^*, \boldsymbol{z}^*] \rho_{\boldsymbol{\pi}}(\boldsymbol{x}^*, \boldsymbol{z}^*) \tag{7}$$

*where subsets $\boldsymbol{X}^* \subseteq \boldsymbol{X}$, $\boldsymbol{Z}^* \subseteq \boldsymbol{O} \setminus \boldsymbol{X}$; and the imitator's occupancy measure $\rho_{\boldsymbol{\pi}}(\boldsymbol{x}^*, \boldsymbol{z}^*)$ is a function of the observational distribution $P(\boldsymbol{O})$ and policy $\boldsymbol{\pi}$.*

Thm. 2 suggests a general procedure to learn an imitating policy via causal IRL. Whenever an identifiable scope $\mathcal{S}$ is found, the identification formula in Eq. (7) permits one to reduce the optimization problem in Eq. (1) to the canonical equation in Eq. (3). One could thus obtain an imitating policy $\boldsymbol{\pi} \sim \mathcal{S}$ by solving Eq. (3) where the expert's occupancy measure $\rho(\boldsymbol{x}^*, \boldsymbol{z}^*) = P(\boldsymbol{x}^*, \boldsymbol{z}^*)$ and the imitator's occupancy measure $\rho_{\boldsymbol{\pi}}(\boldsymbol{x}^*, \boldsymbol{z}^*)$ is given by Eq. (7). As an example, consider the frontdoor diagram described in Fig. 2a and a policy scope $\mathcal{S} = \{\langle X, \emptyset \rangle\}$. The expected reward $\mathbb{E}[Y \mid do(\pi)] = \sum_{x'} \mathbb{E}[Y \mid do(x')] \pi(x')$ and $\mathbb{E}[Y \mid do(x')]$ is identifiable from $P(X, Y, Z)$ using the frontdoor adjustment formula (Pearl, 2000, Thm. 3.3.4). The expected reward $\mathbb{E}[Y \mid do(\pi)]$ of any policy $\pi(X)$ could be written as:

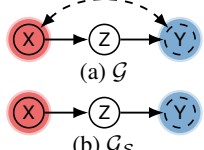

(a) $\mathcal{G}$

(b) $\mathcal{G}_{\mathcal{S}}$

Figure 2: Frontdoor

$$\mathbb{E}[Y \mid do(\pi)] = \sum_{z,x} \mathbb{E}[Y \mid x, z] P(x) \sum_{x'} P(z \mid x') \pi(x'). \tag{8}$$

Let occupancy measures $\rho(x, z) = P(x, z)$ and $\rho_{\boldsymbol{\pi}}(x, z) = P(x) \sum_{x'} P(z \mid x') \pi(x')$. We could thus learn an imitating policy in the frontdoor diagram by solving the canonical equation given by:

$$\nu^* = \min_{\pi \sim \mathcal{S}} \max_{r \in \mathscr{R}} \sum_{x,z} r(x, z) \left( \rho(x, z) - \rho_{\boldsymbol{\pi}}(x, z) \right), \tag{9}$$

where $\mathscr{R}$ is a hypothesis class of the reward function $r(x, z) \triangleq \mathbb{E}[Y \mid x, z]$. The solution $\pi^*(X)$ is an imitating policy performing at least as well as the expert's behavior policy if the gap $\nu^* \leq 0$.

Next, we will describe how to obtain the identification formula in Eq. (7) provided with an identifiable scope $\mathcal{S}$. Without loss of generality, we will assume that the reward $Y$ is the only endogenous variable that is latent in the causal diagram $\mathcal{G}$, i.e., $\boldsymbol{V} = \boldsymbol{O} \cup \{Y\}$.[*] We will utilize a special type of clustering of nodes in the causal diagram $\mathcal{G}$, called the *confounded component* (for short, c-component).

**Definition 6** (C-component (Tian & Pearl, 2002))**.** For a causal diagram $\mathcal{G}$, a subset $\boldsymbol{C} \subseteq \boldsymbol{V}$ is a c-component if any pair $V_i, V_j \in \boldsymbol{C}$ is connected by a bi-directed path in $\mathcal{G}$.

For instance, the frontdoor diagram in Fig. 2a contains two c-components $\boldsymbol{C}_1 = \{X, Y\}$ and $\boldsymbol{C}_2 = \{Z\}$. We will utilize a sound and complete procedure IDENTIFY (Tian, 2002; 2008) for identifying causal effects $\mathbb{E}[Y \mid do(\boldsymbol{\pi})]$ of an arbitrary policy $\boldsymbol{\pi} \sim \mathcal{S}$. Particularly, IDENTIFY takes as input the causal diagram $\mathcal{G}$, a reward $Y$, and a policy scope $\mathcal{S}$. It returns an identification formula for $\mathbb{E}[Y \mid do(\boldsymbol{\pi})]$ from $P(\boldsymbol{O}, Y)$ if expected rewards of all policies $\boldsymbol{\pi} \sim \mathcal{S}$ are identifiable. Otherwise, IDENTIFY$(\mathcal{G}, Y, \mathcal{S}) = $ "FAIL". Details of IDENTIFY are shown in (Zhang et al., 2020, Appendix B). Recall that $\mathcal{G}_{\mathcal{S}}$ is the causal diagram of submodel $M_{\boldsymbol{\pi}}$ induced by policy $\boldsymbol{\pi} \sim \mathcal{S}$. Fig. 2b shows diagram $\mathcal{G}_{\mathcal{S}}$ obtained from the frontdoor graph $\mathcal{G}$ and scope $\mathcal{S} = \{\langle X, \emptyset \rangle\}$ described in Fig. 2a. Let $\boldsymbol{Z}_Y = An(Y)$ be ancestors of $Y$ in $\mathcal{G}_{\mathcal{S}}$. Our next result shows that IDENTIFY$(\mathcal{G}, Y, \mathcal{S})$ is ensured to find an identification formula of the form in Eq. (7) when it is identifiable.

**Lemma 1.** *Given a causal diagram $\mathcal{G}$, a policy scope $\mathcal{S}$ is identifiable from $P(\boldsymbol{O}, Y)$ in $\mathcal{G}$ if and only if IDENTIFY$(\mathcal{G}, Y, \mathcal{S}) \neq$ "FAIL". Moreover, IDENTIFY$(\mathcal{G}, Y, \mathcal{S})$ returns an identification formula of the form in Eq. (7) where $\boldsymbol{X}^* = Pa(\boldsymbol{C}_Y) \cap \boldsymbol{X}$ and $\boldsymbol{Z}^* = Pa(\boldsymbol{C}_Y) \setminus (\{Y\} \cup \boldsymbol{X})$; and $\boldsymbol{C}_Y$ is a c-component containing reward $Y$ in subgraph $\mathcal{G}_{[An(\boldsymbol{Z}_Y)]}$.*

---

[*] Otherwise, one could always simplify the diagram $\mathcal{G}$ and project other latent variables $\boldsymbol{L} \setminus \{Y\}$ using the projection algorithm (Tian, 2002, Sec. 4.5), without affecting the identifiability of target query $E[Y \mid do(\boldsymbol{\pi})]$.

For example, for the frontdoor diagram $\mathcal{G}$ in Fig. 2a, the manipulated diagram $\mathcal{G}_\mathcal{S}$ with scope $\mathcal{S} = \{\langle X, \emptyset \rangle\}$ is described in Fig. 2b. Since $\boldsymbol{Z}_Y = An(Y)_{\mathcal{G}_\mathcal{S}} = \{X, Z, Y\}$, $\boldsymbol{C}_Y$ is thus given by $\{X, Y\}$. Lem. 1 implies that $\boldsymbol{X}^* = Pa(\{X, Y\}) \cap \{X\} = \{X\}$ and $\boldsymbol{Z}^* = Pa(\{X, Y\}) \setminus \{X, Y\} = \{Z\}$. Applying IDENTIFY$(\mathcal{G}, Y, \{\langle X, \emptyset \rangle\})$ returns the frontdoor adjustment formula in Eq. (8).

## 3.1 SEARCHING FOR IDENTIFIABLE POLICY SCOPES

The remainder of this section describes an effective algorithm to find identifiable policy scopes $\mathcal{S}$ had the latent reward signal $Y$ been observed. Let $\mathbb{S}$ denote the collection of all identifiable policy scopes $\mathcal{S}$ from distribution $P(\boldsymbol{O}, Y)$ in the causal diagram $\mathcal{G}$. Our algorithm LISTIDSCOPE, described in Alg. 1, enumerates elements in $\mathbb{S}$. It takes as input a causal diagram $\mathcal{G}$, a reward signal $Y$, and subsets $\boldsymbol{L} = \emptyset$ and $\boldsymbol{R} = \bigcup_{i=1}^n \boldsymbol{PA}_i^*$. More specifically, LISTIDSCOPE maintains two scopes $\mathcal{S}_l \subseteq \mathcal{S}_r$ (Step 2). It performs backtrack search to find identifiable scopes $\mathcal{S}$ in $\mathcal{G}$ such that $\mathcal{S}_l \subseteq \mathcal{S} \subseteq \mathcal{S}_r$. It aborts branches that either (1) all subscopes in $\mathcal{S}_r$ are identifiable (Step 3); or (2) all subscopes containing $\mathcal{S}_l$ are non-identifiable (Step 6). The following proposition supports our aborting criterion.

**Lemma 2.** *Given a causal diagram $\mathcal{G}$, for policy scopes $\mathcal{S}' \subseteq \mathcal{S}$, $\mathcal{S}'$ is identifiable from distribution $P(\boldsymbol{O}, Y)$ in $\mathcal{G}$ if $\mathcal{S}$ is identifiable from $P(\boldsymbol{O}, Y)$ in $\mathcal{G}$.*

At Step 7, LISTIDSCOPE picks an arbitrary variable $V$ that is included in input covariates $\boldsymbol{R}$ but not in $\boldsymbol{L}$. It then recursively returns all identifiable policy scopes $\mathcal{S}$ in $\mathcal{G}$: the first recursive call returns scopes taking $V$ as an input for some actions $X_i \in \boldsymbol{X}$ and the second call return all scopes that do not consider $V$ when selecting values for all actions $\boldsymbol{X}$. We say a policy $\pi$ is associated with a collection of policy scopes $\mathbb{S}$, denoted by $\pi \sim \mathbb{S}$, if there exists $\mathcal{S} \in \mathbb{S}$ so that $\pi \sim \mathcal{S}$. It is possible to show that LISTIDSCOPE produces a collection of identifiable scopes that is sufficient for the imitation task.

---

**Algorithm 1:** LISTIDSCOPE

1: **Input:** $\mathcal{G}, Y$ and subsets $\boldsymbol{L} \subseteq \boldsymbol{R}$
2: **Output:** a set of identifiable policy scopes $\mathbb{S}$
3: Let scopes $\mathcal{S}_r = \{\langle X_i, \boldsymbol{R} \cap \boldsymbol{PA}_i^* \rangle\}_{i=1}^n$ and $\mathcal{S}_l = \{\langle X_i, \boldsymbol{L} \cap \boldsymbol{PA}_i^* \rangle\}_{i=1}^n$.
4: **if** IDENTIFY$(\mathcal{G}, Y, \mathcal{S}_r) \neq$ "FAIL" **then**
5:     Output $\mathcal{S}_r$.
6: **end if**
7: **if** IDENTIFY$(\mathcal{G}, Y, \mathcal{S}_l) \neq$ "FAIL" **then**
8:     Pick an arbitrary $V \in \boldsymbol{R} \setminus \boldsymbol{L}$.
9:     LISTIDSCOPE$(\mathcal{G}, Y, \boldsymbol{L} \cup \{V\}, \boldsymbol{R})$.
10:    LISTIDSCOPE$(\mathcal{G}, Y, \boldsymbol{L}, \boldsymbol{R} \setminus \{V\})$.
11: **end if**

---

**Theorem 3.** *For a causal diagram $\mathcal{G}$ and a reward $Y$, LISTIDSCOPE$(\mathcal{G}, Y, \emptyset, \bigcup_{i=1}^n \boldsymbol{PA}_i^*)$ enumerates a subset $\mathbb{S}^* \subseteq \mathbb{S}$ so that for any $\pi \sim \mathbb{S}$, there is $\pi^* \sim \mathbb{S}^*$ where $\mathbb{E}[Y \mid do(\pi)] = \mathbb{E}[Y \mid do(\pi^*)]$.*

Moreover, LISTIDSCOPE outputs identifiable policy scopes with a polynomial delay. This follows from the observation that LISTIDSCOPE searches over a tree of policy scopes with height at most $|\bigcup_{i=1}^n \boldsymbol{PA}_i^*|$ and IDENTIFY$(\mathcal{G}, Y, \mathcal{S})$ terminates in polynomial steps w.r.t. the size of diagram $\mathcal{G}$.

## 4 EXPERIMENTS

In this section, we demonstrate our framework on various imitation learning tasks, ranging from synthetic causal models to real-world datasets, including highway driving (Krajewski et al., 2018) and images (LeCun, 1998). We find that our approach is able to incorporate parametric knowledge about the reward function and achieve effective imitating policies across different causal diagrams. For all experiments, we evaluate our proposed `Causal-IRL` based on the *canonical equation* formulation in Eq. (3). As a baseline, we also include: (1) standard `BC` mimicking the expert's nominal behavior policy; (2) standard `IRL` utilizing all observed covariates preceding every $X_i \in \boldsymbol{X}$ while being blind to causal relationships in the underlying model; and (3) `Causal-BC` (Zhang et al., 2020; Kumor et al., 2021) that learn an imitating policy with the sequential $\pi$-backdoor criterion. We refer readers to (Ruan et al., 2023, Appendix D) for additional experiments and more discussions on the experimental setup.

**Backdoor** Consider an SCM instance compatible with Fig. 1c including binary observed variables $Z_1, X_1, Z_2, X_2, Y \in \{0, 1\}$. `Causal-BC` utilizes a sequential $\pi$-backdoor admissible scope $\{\langle X_1, \{Z_1\} \rangle, \langle X_2, \{Z_2\} \rangle\}$; while `Causal-IRL` utilizes the scope $\{\langle X_1, \emptyset \rangle, \langle X_2, \{Z_2\} \rangle\}$ satisfying the minimal sequential $\pi$-backdoor. Simulation results, shown in Fig. 3a, reveal that `Causal-IRL` consistently outperforms the expert's policy and other imitation strategies by exploiting additional parametric knowledge about the expected reward $\mathbb{E}[Y \mid X_1, X_2, Z_2]$; `Causal-BC` is able to achieve the expert's performance. Unsurprisingly, neither `BC` nor `IRL` is able to obtain an imitating policy.

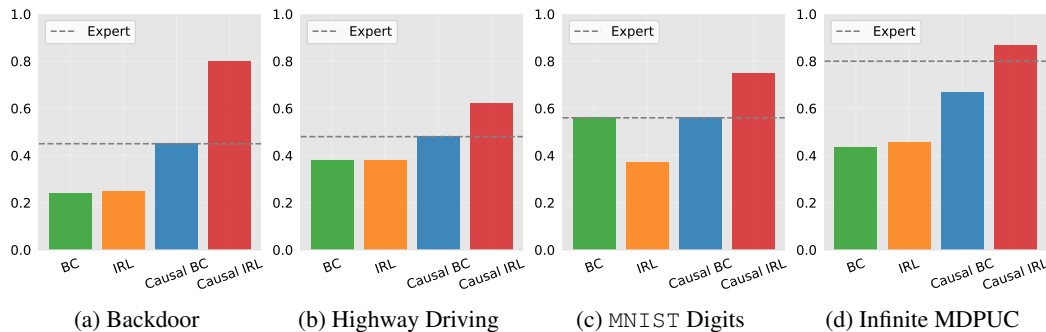

Figure 3: Simulation results (a, b, c, d) for our experiments, where y-axis represents the expected reward of learned policies in the actual causal model; the grey dashed line denotes the expert's reward.

**Highway Driving** We consider a learning scenario where the agent learns a driving policy from the observed trajectories of a human expert. Causal diagram of this example is provided in (Ruan et al., 2023, Appendix D, Fig. 4) where $X_1$ is the accelerations of the ego vehicle at the previous step; $Z_1$ is both longitudinal and lateral historical accelerations of the ego vehicle two steps ago; $X_2$ is the velocity of the ego vehicle; $Z_2$ is the velocity of the preceding vehicle; $W$ indicates the information from surrounding vehicles. Values of $X_1, X_2, Z_1, Z_2$ are drawn from a real-world driving dataset `HighD` Krajewski et al. (2018). The reward $Y$ is decided by a non-linear function $f_Y(X_2, Z_2, U_Y)$. Both `Causal-IRL` and `Causal-BC` utilize the scope $\{\langle X_1, \emptyset \rangle, \langle X_2, \{Z_2\} \rangle\}$. `Causal-IRL` also exploits the additional knowledge that the expected reward $\mathbb{E}[Y \mid X_1, X_2, Z_2]$ is a monotone function via reward augmentation (Li et al., 2017). Simulation results are shown in Fig. 3b. We found that `Causal-IRL` performs the best among all strategies. `Causal-BC` is able to achieve the expert's performance. `BC` and `IRL` perform the worst among all and fail to obtain an imitating policy.

**MNIST Digits** Consider again the frontdoor diagram in Fig. 2a. To evaluate the performance of our proposed approach in high-dimensional domains, we now replace variable $Z$ with sampled images drawn from `MNIST` digits dataset (LeCun, 1998). The reward $Y$ is decided by a linear function taking $Z$ and an unobserved confounder $U_{X,Y}$ as input. The `Causal-IRL` formulates the imitation problem as a two-person zero-sum game through the frontdoor adjustment described in Eq. (9), which can be solved by the MW algorithm (Freund & Schapire, 1999; Syed & Schapire, 2008). As shown in Fig. 3c, simulation results reveal that `Causal-IRL` outperforms `Causal-BC` and `BC`; while `IRL` performs the worst among all the algorithms.

**Infinite MDPUC** To demonstrate our proposed framework in the sequential decision-making setting with an infinite horizon, we consider a generalized Markov decision process incorporating unobserved confounders (Ruan & Di, 2022), called the MDPUC (Zhang & Bareinboim, 2022). This sequential model simulates real-world driving dynamics. By exploiting the Markov property over time steps, we are able to decompose the causal diagram over the infinite horizon into a collection of sub-graphs, one for each time step $i = 1, 2, \ldots$. Fig. 1d shows the causal diagram spanning time steps $i = 1, 2, 3$. As a comparison, `BC` and `IRL` still utilize the stationary policy $\{\langle X_i, \{Z_i\} \rangle\}$. By applying Thm. 1 at each time step, we obtain a $\pi$-backdoor admissible policy scope $\{\langle X_i, \{Z_i, X_{i-1}, Z_{i-1}\} \rangle\}$ for `Causal-IRL` and `Causal-BC`. Simulation results are shown in Fig. 3d. One could see by inspection that `Causal-IRL` performs the best and achieves the expert's performance.

## 5 CONCLUSION

This paper investigates imitation learning via inverse reinforcement learning (IRL) in the semantical framework of structural causal models. The goal is to find an effective imitating policy that performs at least as well as the expert's behavior policy from combinations of demonstration data, qualitative knowledge the data-generating mechanisms represented as a causal diagram, and quantitative knowledge about the reward function. We provide a graphical criterion (Thm. 1) based on the sequential backdoor, which allows one to obtain an imitating policy by solving a canonical optimization equation of causal IRL. Such a canonical formulation addresses the challenge of the presence of unobserved confounders (UCs), and is solvable by leveraging standard IRL algorithms (Props. 1 and 2). Finally, we move beyond the backdoor criterion and show that the canonical equation is achievable whenever expected rewards of policies are identifiable had the reward also been observed (Thms. 2 and 3).

ACKNOWLEDGEMENTS

This research was supported in part by the NSF, ONR, AFOSR, DoE, Amazon, JP Morgan, and The Alfred P. Sloan Foundation.

ETHICS STATEMENT

This paper investigates the theoretical framework of causal inverse RL from the natural trajectories of an expert demonstrator, even when the reward signal is unobserved. Input covariates used by the expert to determine the original values of the action are unknown, introducing unobserved confounding bias in demonstration data. Our framework may apply to various fields in reality, including autonomous vehicle development, industrial automation, and chronic disease management. A positive impact of this work is that we discuss the potential risk of training IRL policy from demonstrations with the presence of unobserved confounding (UC). Our formulation of causal IRL is inherently robust against confounding bias. For example, solving the causal IRL problem in Eq. (1) requires the imitator to learn an effective policy that maximizes the reward in a worst-case causal model where the performance gap between the expert and imitator is the largest possible. More broadly, automated decision systems using causal inference methods prioritize safety and robustness during their decision-making processes. Such requirements are increasingly essential since black-box AI systems are prevalent, and our understandings of their potential implications are still limited.

REPRODUCIBILITY STATEMENT

The complete proof of all theoretical results presented in this paper, including Thms. 1 and 2, is provided in (Ruan et al., 2023, Appendix B). Details on the implementation of the proposed algorithms are included (Ruan et al., 2023, Appendix C). Finally, (Ruan et al., 2023, Appendix D) provides a detailed description of the experimental setup. Readers could find all appendices as part of the supplementary text after "References" section. We provided references to all existing datasets used in experiments, including HIGHD (Krajewski et al., 2018) and MNIST (LeCun, 1998). Other experiments are synthetic and do not introduce any new assets. Source codes for all experiments and simulations are released in the complete technical report (Ruan et al., 2023).

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
