# OpenReview forum: "Causal Imitation Learning via Inverse Reinforcement Learning"
_ICLR.cc/2023/Conference — ICLR 2023 poster_

### Official Review · Reviewer_o8Xw · 2022-10-25

**Confidence:** 3
**Correctness:** 3
**Technical Novelty And Significance:** 2
**Empirical Novelty And Significance:** 2
**Recommendation:** 6

**Clarity, Quality, Novelty And Reproducibility:**


**Clarity**
The presentation is clear.

**Quality**
This paper is of high quality.

**Novelty**
The proposed method in this paper is novel.

**Reproducibility**
Details of the algorithms and experiments have been provided.


**Strength And Weaknesses:**

## Strength

+ The proposed method is novel. Under the structural causal model of inverse reinforcement learning, this paper extends the graphical condition for causal behavioral cloning to causal inverse reinforcement learning. This was done by formulating the problem as learning to paly a two-player zero-sum game, where the prior knowledge about the latent rewards can be incorporated. With informative knowledge, the imitator can outperform the expert in causal environment.

+ The proposed method is well motivated with theoretical analysis, and naturally extended to two existing inverse reinforcement learning algorithms, i.e. MWAL and GAIL. The proposed method was well supported by the empirical results.

## Weakness

+ More diverse and challenging experiments are expected.






**Summary Of The Paper:**

This paper studies imitation learning through the lens of structural causal models. Specifically, this paper proposes graphical conditions to enable imitator to learn an expert-level policy, when unobserved confounders (UCs) occur. Interestingly, the proposed method can extends to existing IRL algorithms when UCs are present. Two showcased algorithms are causal version of multiplicative weights algorithm (MWAL) and generative adversarial imitation learning (GAIL). The proposed methods have been evaluated using real-world and synthetic data, showing supervior performance than the original ones.

**Summary Of The Review:**

Overall, this paper provides a novel causal inverse reinforcement learning method. The quality is good.

---

> ### Author Response · Authors · 2022-11-13
> **Response to Reviewer o8Xw**
>
> We are glad that the reviewer found the proposed method is “novel” and “well motivated with theoretical analysis.” We think that a theoretical understanding of IRL based on causality is critical for agents to learn to make decisions when unobserved confounders exist, and we hope that the clarification below helps to clarify our contributions and experiments further.
>
>
> ---
> > #### “weakness: More diverse and challenging experiments are expected.”
>
> We thank the review for the suggestion and would like to take this opportunity to elaborate further on comprehensiveness of our experiments. Further details of the experimental setups and additional experiments could be found in Appendix D (Page 20-25). We have validated our framework through extensive systematic experiments covering different dimensions of the causal imitation learning tasks. These dimensions include:
>
>  (1) causal assumptions, e.g., backdoor graphs and frontdoor graphs;
>
>  (2) parametric families of reward functions, e.g., linear and nonlinear reward functions;
>
>  (3) multiple datasets, including the high-dimensional domains (MNIST), and real-world trajectories of human driving on the highway (HighD).
>
> (4) finite v.s. Infinite horizons: We not only evaluate our algorithm on causal diagrams with finite horizons, but also on the long-sequence decision problem with infinite horizons, e.g., the MDPUC model with infinite horizons.
>
> Note that in all those experiments, the expert’s demonstrations are contaminated with biases induced by unobserved confounders (UCs). Therefore, they are non-trivial for existing inverse RL algorithms if applied directly. Our proposed augmentation procedure allows these algorithms to obtain an effective policy that imitates the expert’s performance (if possible), even when demonstrations are imperfect and UCs generally exist.
>
> Also, most benchmarks for imitation learning are based on the Markov decision process model, which does not explicitly consider the presence of UCs. In these cases, our framework generally coincides with the classical IRL when UCs do not exist. In this paper, we evaluate our framework using other commonly used benchmarks in Causal Imitation Learning (Zhang et al., 2020; Etesami et al., 2020; Kumor et al., 2021), e.g., HighD. In summary, we believe that our experiments are comprehensive, covering a wide range of imitation settings, and capturing the challenges of the unobserved confounding, which is the main goal of this paper. Also, we note this is not the last but the first paper that tackles the challenges of inverse RL whenever unobserved confounders are present, noting that this is a quite pervasive phenomenon present in most practical settings in the real world.
>
>
> ---
> > #### “The contributions are only marginally significant or novel.”
>
> We would like to highlight the main contributions of this paper further since we respectfully disagree with this point. This paper aims to extend existing IRL approaches to more generalized settings where the expert’s demonstrations are imperfect, and unobserved confounding exists. Our idea is to utilize causal relationships embedded in the environment so that the learner can address the bias/distribution drift due to unobserved confounding in demonstration data. To further position this work, we summarize the current literature on imitation learning as follows, in order to highlight that our work fills a critical literature gap where unobserved confounders exist in the sequential imitating processes and the expert is sub-optimal::
>
> |     | Optimal Expert | Sub-optimal Expert|
> | --- | ----------- |  ----------- |
> | Unconfounded | Behavior Cloning | Inverse RL |
> | Confounded | Causal BC | Causal IRL (**our work**)  |
>
> To sum up, behavior cloning (BC) is able to achieve satisfactory performance when the demonstration data is unconfounded, and the expert is (near) optimal. Inverse RL (IRL) can outperform the suboptimal expert by exploiting additional parametric knowledge about the reward signal. However, both BC and IRL are fragile to data drift due to the presence of unobserved confounders. Causal BC (Zhang et al., 2020) produces a confounding-robust policy but is still limited by the performance of a suboptimal expert. Finally, this paper empowers IRL methods with the causal inference theory so that the IRL imitator could produce a policy robust to both unobserved confounding and the suboptimal expert. Given the significance of imitation learning and the prevalence of challenges of imperfect and biased demonstrations, we are confident that our paper would have a broader impact across disciplines in artificial intelligence.

---

### Official Review · Reviewer_Ne7b · 2022-10-26

**Confidence:** 3
**Correctness:** 4
**Technical Novelty And Significance:** 3
**Empirical Novelty And Significance:** 3
**Recommendation:** 8

**Clarity, Quality, Novelty And Reproducibility:**

Clarity and quality: Both extremely high
Novelty: While these ideas have been "in the water" in the community for a while, this work certainly represents a novel synthesis, and provides the first algorithm that can convincingly exceed expert performance in scenarios with UCs.
Reproducibility: Probably possible by an extremely patient researcher, but the authors should strongly consider open sourcing.

**Strength And Weaknesses:**

Strengths: The submission does a fantastic job laying the groundwork for causal reinforcement learning, motivating its algorithms, theorems, and results within the formalisms of structured causal models.   I also greatly appreciated the extensive FAQ in the appendix.  The experiments clearly demonstrate the power of the method.

Weakness: While the exposition of the paper is exceptionally high quality, there does not appear to be a code repo associated with the submission.  This method, and its uptake by the community, would greatly benefit from open sourcing of code / experiments.

**Summary Of The Paper:**

The authors propose a new inverse reinforcement learning algorithm that's able to provably meet the performance of an expert demonstrator in the presence of uncontrolled confounders, when certain conditions are met (i.e., does the agent have the causal structure of the data generating process right?).

**Summary Of The Review:**

An excellent submission to the study of causal reinforcement learning.  Please open source it :)

---

> ### Author Response · Authors · 2022-11-13
> **Response to Reviewer Ne7b**
>
> Thank you for taking the time to read the paper and for the positive assessment, and acknowledging that our work “does a fantastic job laying the groundwork for causal reinforcement learning”.
>
> ---
> > #### “Weakness: While the exposition of the paper is exceptionally high quality, there does not appear to be a code repo associated with the submission. This method, and its uptake by the community, would greatly benefit from open sourcing of code / experiments.”
>
> We agree that “open-sourcing codes/experiments” is beneficial to the community. We will release our codes with the updated manuscript, thank you for the suggestion.
>
>
> ---
> > #### “does the agent have the causal structure of the data generating process right?”
>
> Yes, we assume the agent has access to structural assumptions about the data-generating process represented as a causal diagram $\mathcal{G}$. Still, we do not assume that it has access to the particular instantiation of the structural causal model (SCM), which is much stronger than the specific causal diagram. Also, there exist quite general and systematic methods under the rubrics of “causal discovery” capable of learning a causal diagram (or its equivalence class) from observational (Spirtes et al., 2000) and experimental data (Jaber et al., 2019). This means that the dependence on domain experts could be minimized.

---

### Official Review · Reviewer_iWpt · 2022-10-27

**Confidence:** 3
**Correctness:** 3
**Technical Novelty And Significance:** 3
**Empirical Novelty And Significance:** 2
**Recommendation:** 6

**Clarity, Quality, Novelty And Reproducibility:**

The proposed work has its value especially for imitation learning. However, the paper lacks a clear problem formulation. The presentation, in particular, the use of variables, is very confusing and hard to follow. The solution part does not appear to be complete. A number of concepts are not clearly defined.

For example, variable X and x used to refer to different concepts in Section 1.1 and Section 2. The concept of "intervention" is not clearly defined and denoted using different variables (e.g.,  X and do(\pi)).

**Strength And Weaknesses:**

Strengths:

1. The problem described in the paper is meaningful and challenging.
2. Theoretical proofs for theorems are provided.

Weaknesses:

1. The paper lacks a clear problem formulation. The input/output and objectives of the problem are not provided.
2. The paper uses many different variables, where some of them are either undefined, or repeatedly used for different purposes. The presentation makes it hard to follow the details of the paper.
3. The paper seems to be incomplete. The only algorithm only shows how to identify identifiable policy scope. However, how to use this algorithm with IRL or GAIL is not provided.
4. The experiments are not convincing. Why would the MNIST dataset be used, and how to learn a policy from this dataset?
5. How is the problem setting in this paper different from a partially observable MDP (POMDP)? More discussions are needed.

**Summary Of The Paper:**

This paper proposes a framework for causal imitation learning. The problem assumes that the experts has access to latent features that are not directly observable to imitators. The solution is based on identifying the minimal \pi-backdoor admissible scope from the causal diagram. Experiments on a few datasets are provided to validate the effectiveness of the proposed approach.

**Summary Of The Review:**

The paper proposes an imitation learning framework, where the imitator can learn policies from observations with unobserved confounding. The solution is based on identifying identifiable policy scope from a causal diagram of the underlying decision process. Experiments on a few datasets are conducted to validate the effectiveness of the proposed solution.

The problem discussed in this paper is interesting, important, and technically challenging. However, paper fail to provide a clear formulation of the problem (input/output/objective, MDP formulation). The presentation, in particular the non-rigorous use of  variables and concepts make it hard to follow the details of the paper. The solution lacks the steps to complete the imitation learning loop. Experiments can be strengthened by using more relevant datasets and evaluating on additional measures (e.g., fidelity of learned polices).

---

> ### Author Response · Authors · 2022-11-13
> **Response to Reviewer iWpt [1/4]**
>
> We appreciate the reviewer’s thoughtful feedback. We believe that a few misreadings of our work made some of the evaluations overly harsh and would ask reviewers to reconsider our paper in the light of the clarifications provided below.
>
> ---
> > #### “1. The paper lacks a clear problem formulation. The input/output and objectives of the problem are not provided.”
>
> We respectfully disagree with this statement and would like to point out that the problem formulation has been provided in Section 2. Particularly, as mentioned on Page 4, Lines 6-13,  “the goal of the agent is to learn an **imitating policy** $\pi^*$ that achieves the expert's performance, $\mathbb{E}_M [Y \mid do(\pi^*)] \geq \mathbb{E}_M[Y].$
> In words, the right-hand side is the expert's performance that the agent wants to achieve, while the left-hand side is the real reward experienced by the agent.
> Throughout this paper, detailed parametrizations of the underlying SCM $M$ are assumed to be **unknown** to the agent. Instead, the agent has access to a causal diagram $\mathcal{G}$ associated with $M$, and observed trajectories of the expert's demonstrations, summarized as the observational distribution $P(\boldsymbol{O})$.”
>
> In other words, the problem of causal IRL can be formulated as:
> - Input: the expert's demonstrations, summarized as the observational distribution $P(\boldsymbol{O})$, and a causal diagram $\mathcal{G}$
> - Output: an imitating policy $\pi^*$
> - Objective: “to learn an imitating policy $\pi^*$ that achieves the expert's performance, $\mathbb{E}_M[Y \mid do(\pi^*)] \geq \mathbb{E}_M[Y]$”.
>
> As an example, consider the causal diagram $\mathcal{G}$ described in Fig. 1(a). In this case, the observational distribution is given by $P(X, Z)$; the reward signal $Y$ is unobserved. The expert’s performance is given by $E[Y]$. Since this model has no unobserved confounder, the agent could learn an imitating policy from the observational distribution as $\pi(X|Z) = P(X|Z)$. It is verifiable that this policy achieves the expert’s performance, i.e.,
> $$
> E[Y|do(\pi)] = \sum_{x, z}E[Y|x, z] \pi(x|z)P(z)
> $$
>
> $$
> =\sum_{x, z}E[Y|x, z] P(x|z) P(z)
> $$
>
> $$
> =E[Y]
> $$
>
> The second step follows from $\pi(X|Z) = P(X|Z)$ and the last step follows from basic probabilistic marginalization.
>
> This paper extends the above example from two critical aspects. First, we consider the generalized settings where the observational data is contaminated with the presence of unobserved confounding, which requires additional adjustment procedures to debias the data. Second, the performance of the cloning policy  $\pi(X|Z) = P(X|Z)$ is bounded by the expert’s performance $E[Y]$. We explore the inverse reinforcement learning approach, which allows the learner to outperform the expert by utilizing parametric knowledge about the reward function (e.g., $E[Y|x, z]$). We hope this answers the reviewer’s question and would be happy to provide additional clarification if there is any confusion about the problem formulation.

---

> > ### Comment · Reviewer_iWpt · 2022-11-14
> > **Reply to responses to my review comments**
> >
> > I would like to thank the authors for providing detailed explanations to my comments. After carefully checking the paper and the appendix again with the help of the authors' responses, I believe the paper makes more sense to me now. In particular the authors have explained the choice of the experiment datasets and how to complete the learning process with alg. 3 and 4.  The revision to the draft also makes the notations and definitions clearer.
> >
> > Since most of my previous comments were about the presentation and organization of the paper, I would be fine with giving a higher score to the paper after the responses/revisions
> >
> > However I would still encourage the authors to improve the presentation to help a broader range of readers better understand their work. For example, (1) adding a table of notations (in the appendix if no space in the main paper), (2) highlighting the problem formulation in the way that was written in the response, and (3) making clear references to Alg. 3 and Alg. 4 in the main paper. The goal of the problem, as suggested by the authors, is to learn an imitating policy \pi*. However, algorithm 1 only finds a policy scope, which is inconsistent with their problem statement. Alg. 3 and 4 are needed to close the loop. In fact there is no output listed in Alg. 1 & 4, which is another presentation flaw. Addressing these comments would further improve the paper.

---

> > > ### Author Response · Authors · 2022-11-16
> > > **Thanks for your suggestions.**
> > >
> > > We summarize below how we have updated the manuscript to address the reviewer’s suggestions. All changes are highlighted in the rebuttal revision. Hopefully without seeming to pander, please let us note that we believe that this process has substantially improved the manuscript; thank you.
> > >
> > > ---
> > > > #### “adding a table of notations (in the appendix if no space in the main paper).”
> > >
> > > We added Table 2 in Appendix F, summarizing important notations used in this paper.
> > >
> > > ---
> > > > #### “highlighting the problem formulation in the way that was written in the response.”
> > >
> > > We added a formal definition of imitating policy in Definition 1 (Page 4), which highlights the goal of the paper. We also highlighted the input and output of the agent on Page 4, Lines 2-4. Finally, we added the problem formulation of causal IRL in Q13, Appendix F, as written in the response.
> > >
> > > ---
> > > > #### “making clear references to Alg. 3 and Alg. 4 in the main paper. … Alg. 3 and 4 are needed to close the loop.”
> > >
> > > We added a clear reference to Alg. 3 and Alg. 4 in the main manuscript. On Page 6, it states “In this paper, we solve for an imitating policy $\pi^*$ in Eq. (3) using state-of-the-art IRL algorithms, provided with common choices of parametric reward functions. These algorithms include the multiplicative-weights algorithm (MWAL) (Syed & Schapire, 2008) and the generative adversarial imitation learning (GAIL) (Ho & Ermon, 2016). We refer readers to Algs. 3 and 4 in Appendix C for more discussion on the pseudo-code and implementation details.” We clarified that once the augmentation procedure is performed, our proposed approach computes an imitating policy using existing IRL algorithms including MWAL and GAIL.
> > >
> > > ---
> > > > #### “there is no output listed in Alg. 1 & 4, which is another presentation flaw.”
> > >
> > > We added output statements for Algs 1-4. Particularly, Alg. 1 returns “a set of identifiable policy scopes $\mathbb{S}$”; Alg. 2 returns “an expression for $Q[\boldsymbol{C}]$ in terms of $Q[\boldsymbol{T}]$ or fail to determine”; Both Alg. 3 and Alg. 4 return “an imitating policy $\pi^*$.”
> > >
> > > ---
> > >
> > > After all, we hope these changes follow what you had in mind, and will be happy to refine the text further. Thank you again for the suggestions.
> > >
> > > Authors of Paper #3199

---

> ### Author Response · Authors · 2022-11-13
> **Response to Reviewer iWpt [2/4]**
>
> ---
> > #### “2. The paper uses many different variables, where some of them are either undefined, or repeatedly used for different purposes. The presentation makes it hard to follow the details of the paper. …
> > #### “A number of concepts are not clearly defined…”
> > #### “For example, variable X and x used to refer to different concepts in Section 1.1 and Section 2. The concept of "intervention" is not clearly defined and denoted using different variables (e.g., X and do(\pi)).”
>
>
> We would like to point out respectfully that the notational confusion pointed out by the reviewer was actually stressed in the paper. As mentioned in Page 3, Line 1: “We use capital letters to denote random variables ($X$) and small letters for their values ($x$). For a set $\boldsymbol{X}$, let $\lvert \boldsymbol{X} \rvert$ denote its dimension. The probability distribution over variables $\boldsymbol{X}$ is denoted by $P(\boldsymbol{X})$.”
>
> An agent interacts with the underlying environment by performing interventions (i.e., actions). In other words, the concept of “interventions” in causal inference is “actions” in reinforcement learning or imitation learning. In this paper, we consider two types of interventions, including the atomic intervention $do(\boldsymbol{x})$ and the policy intervention $do(\pi)$. Both interventions have been well-defined in the literature, in the manuscript, and will be further elaborated on here. Page 2, Lines 14-16, states, “an intervention on a subset $\boldsymbol{X} \subseteq \boldsymbol{V}$, denoted by $do(\boldsymbol{x})$, is an operation where values of $\boldsymbol{X}$ are set to constants $\boldsymbol{x}$, replacing the functions $f_{\boldsymbol{X}} = \{f_{X}: \forall X \in \boldsymbol{X}\}$ that would normally determine their values.” Similarly, “an intervention on actions $X$ following a policy $\pi$, denoted by $do(\pi)$, entails a submodel $M_{\pi}$ from a SCM $M$ where the expert's policy $f_{X}$ are replaced with decision rules $X_i \sim \pi_i(X_i \mid Z_i)$ for every $X_i \in \boldsymbol{X}$.” We have updated the manuscript to highlight these definitions further.
>
> We believe that this paper is self-contained and all notations are properly defined.  If the reviewer still has questions about our revised manuscript, we would be happy to provide further elaboration.
>
>
> ---
> > #### “3. The paper seems to be incomplete. The only algorithm only shows how to identify identifiable policy scope. However, how to use this algorithm with IRL or GAIL is not provided.”
>
> We thank the reviewer for the suggestion and would like to take this opportunity to elaborate on how the algorithm of identifying policy scope is incorporated into causal MWAL/GAIL.
>
> LISTIDSCOPE “describes an effective algorithm to find identifiable policy scopes $\mathcal{S}$ had the latent reward signal $Y$ been observed.” Theorem 2 ”shows that whenever an identifiable policy scope S is found, one could always reduce the causal IRL problem to the canonical optimization equation in Eq. (4).” (Page 7, Lines 2-4). Finally, Props 1 and 2 describe how to solve the canonical IRL equation in Eq (4) using standard MWAL and GAIL algorithms. “We refer readers to Appendix C for more discussion on the pseudo-code and implementation details.” Particularly, detailed procedures of Causal MWAL and Causal GAIL are provided in Alg. 3 and Alg. 4, respectively.

---

> ### Author Response · Authors · 2022-11-13
> **Response to Reviewer iWpt [3/4]**
>
> ---
> > #### “4.1. Why would the MNIST dataset be used, and how to learn a policy from this dataset?“
>
> Most benchmarks for imitation learning are based on the Markov decision process model, which does not explicitly consider the presence of unobserved confounders (UCs). In these cases, our framework generally coincides with the classical IRL when UCs do not exist. In this paper, we evaluate our framework using other commonly used benchmarks in Causal Imitation Learning (Zhang et al., 2020; Etesami et al., 2020; Kumor et al., 2021), e.g., MNIST and highD. Particularly, in the frontdoor diagram used in Experiment 3, “we now replace variable $Z$ with sampled images drawn from MNIST digits dataset.” Here, variable $Z$ is a mediator intercepting all causal relationships between the action $X$ and the outcome $Y$. Doing so allows us to evaluate our proposed approach in a more complex and high-dimensional domain (than discrete instances/domains).
>
> As for the policy learning, Appendix D states, “for discrete domains (Experiments 1, 5, 6 and 7), we estimate transition distributions using empirical means and solve for an optimal policy in the RL step using dynamic programming. For an environment with continuous states and actions space (Experiments 2, 3, 4 and 8), we solve for the optimal policy using a stochastic policy gradient (Sutton et al., 1999). Expected rewards of policies are estimated by a Monte-Carlo method computing empirical means over the agent’s trajectories. Effective function approximation methods also exist to evaluate the value function of policies in high-dimensional domains (Tsitsiklis & Van Roy, 1997; Murphy, 2005). “
>
> ---
> > #### “4.2. The experiments are not convincing. … Experiments can be strengthened by using more relevant datasets and evaluating on additional measures (e.g., fidelity of learned policies).”
>
> We thank the reviewer for the suggestion and would like to take this opportunity to elaborate further on the comprehensiveness of our experiments. We believe that our experiments are extensive since they cover a diverse range of imitation settings and capture the challenges of unobserved confounding, which is the main goal of this paper.
>
> Details of the experimental setups and additional experiments can be found in Appendix D. We have validated our framework through extensive systematic experiments, covering different dimensions of the causal imitation learning tasks. These dimensions include: causal assumptions, e.g., backdoor graphs and frontdoor graphs; parametric families of reward functions, e.g., linear and nonlinear reward functions; and multiple datasets, including the high-dimensional domains (MNIST), and real-world trajectories of human driving on the highway (HighD). We also evaluate our algorithm on the long-sequence decision problem with infinite horizons, e.g., the MDPUC model with infinite horizons. Note that in all those experiments, the expert’s demonstrations are contaminated with biases induced by unobserved confounders (UCs). Therefore, they are non-trivial for existing inverse RL algorithms if applied directly. Our proposed augmentation procedure allows these algorithms to obtain an effective policy that imitates the expert’s performance (if possible), even when demonstrations are imperfect and UCs generally exist.
>
> Finally, we evaluate our algorithms with measures that are commonly used in the imitation learning literature. The average rewards of candidate algorithms and variances with respect to the random seed are reported in Table 1, Appendix D. Note that in most of our experiments, the demonstrator is suboptimal and does not always select the best possible action. The goal of our proposed algorithm is to produce a policy that consistently dominates the demonstrator, instead of naively cloning the demonstrator’s policy. Therefore, the fidelity of learned policies is not representative of the agent’s performance, thus, not reported.

---

> ### Author Response · Authors · 2022-11-13
> **Response to Reviewer iWpt [4/4]**
>
> ---
> > #### “5. How is the problem setting in this paper different from a partially observable MDP (POMDP)? More discussions are needed.”
>
> This is a good question and we thank the reviewer for the opportunity that allows us to clarify this question further. In POMDP models where the underlying state is only partially observed, it is generally infeasible to learn an effective policy that achieves the expert’s performance from the observational data alone. That is, POMDP models are generally “non-imitable.” Such an observation was formalized in (Theorem 2, Zhang et al., 2020). This means that to obtain an effective imitating policy, one must explore additional assumptions about the underlying environment. Existing approaches often utilize parametric assumptions about the system dynamics, e.g., the unobserved state is discrete and finite, and the cardinality is accessible to the learner.
>
> Our analysis of imitation learning builds on the semantical framework of structural causal models (SCM) (Pearl, 2000). It could be seen as a more flexible class of environmental models, which includes both MDP and POMDP. Compared to MDP, the representation of SCMs allows the presence of UCs, instead of being assumed away. This enables us to investigate the challenges of data drift in the expert’s demonstrations due to unobserved confounding bias. On the other hand, unobserved states in an SCM could be sparse and local, only affecting a subset of variables in the system. This differs from a standard POMDP model, where the latent state could have causal relationships with all observations and actions at every time step. Structural constraints in a SCM could be represented as a directed acyclic causal diagram. By exploiting causal relationships embedded in the environment, we can perform a more fine-grained analysis of imitation learning with the presence of UCs, and provide novel imitation solutions in cases that were not previously solvable.

---

### Official Review · Reviewer_U7KX · 2022-11-06

**Confidence:** 4
**Correctness:** 3
**Technical Novelty And Significance:** 3
**Empirical Novelty And Significance:** 2
**Recommendation:** 6

**Clarity, Quality, Novelty And Reproducibility:**

The paper is generally well-written but some more explanations can be given in the paper. For instance, it is good to discuss the projection algorithm mentioned in the footnote on page 7 and also mention why it is required to perform such projections. Regarding the novelty of the paper, I think the causal formulation of causal IRL and the algorithm LISTIDSCOPE are somewhat novel but it is not clear how causality can help to get better performance than the expert. The example in the introduction is not convincing. It seems that the results can be reproduced based on the explanation in the appendix.

**Strength And Weaknesses:**

Strength:

- The authors provided a nice formulation of inverse RL in the case that we have access to the causal graph. Moreover, they also gave some examples of this formulation for MWAL and GAIL.
- They presented a method that can enumerate all identifiable policy scopes from $P(O,Y)$.

Weakness:

- Regarding the gap $\nu^*$, it is not clear why the authors only considered the cases that $\nu^*\leq 0$. This can only happen if the expert has suboptimal performance in the environment (or in causal language, in some SCM $M$) that is acting.


In the following, I give my detailed comments:
- It would be great if the authors can give some real cases that the performance of an expert is suboptimal even in the environment that is acting in it. The example in the introduction is not convincing enough as it is not clear why the expert acts based on just $X$ which results in poor performance.
- The authors considered a ``sequential decision-making" setting however, it seems that $Y$ is not indexed with time steps. How can the results in the paper be extended to the more general setting where we have a sequence of $Y_i$'s? In the FAQ, it is mentioned that $Y$ can be a set of variables but it might be the case that the cumulative reward is identifiable while each $Y_i$ is not identifiable from $P(O,Y)$.
- Under what conditions, is $\nu^*$ positive?
- What is the exact definition of effective actions and covariates in Theorem 1? What happens to other variables $\mathbf{X}$ and $\mathbf{Z}$ in Theorem 1?



**Summary Of The Paper:**

The authors considered the problem of imitation learning when the underlying causal structure of the environment is given. They provided a causal formulation of the problem of inverse RL (IRL) (equation (2)). They introduced the notion of minimal $\pi$-backdoor admissible scope and showed that the effect of such policies on the average reward can be computed from the observational distribution over $(O,Y)$, i.e., the observable variables and the reward. Based on this, a canonical equation for IRL is given in (4) where for two settings of MWAL and GAIL, this formulation is simplified. Then, the authors used a method called IDENTIFY which is beyond the graphical condition in $\pi$-backdoor admissible scope and it is sound and complete for checking whether a policy scope is identifiable from $P(O,Y)$. Finally, they proposed LISTIDSCOPE to enumerate all identifiable policy scopes.

**Summary Of The Review:**

The submitted paper shows the advantage of considering the underlying causal diagram in the problem of inverse RL and provides a method to enumerate all identifiable policy scopes. Based on this, the canonical equation of causal IRL in (4) can be computed for such policies. The experiments showed that causal IRL can have better performance than the expert. The main ambiguity in the paper is for the case $\nu^*>0$ and it is required to justify why the expert is not acting optimally even at least one SCM $M$.

---

> ### Author Response · Authors · 2022-11-13
> **Response to Reviewer U7KX [1/4]**
>
> We thank the reviewers for the helpful feedback. We hope the clarification below answers your questions and would be happy to provide further elaboration in case there are any remaining concerns.
>
>
>
> ---
> > #### “Weakness: Regarding the gap $\nu^*$, it is not clear why the authors only considered the cases that $\nu^*\leq 0$. This can only happen if the expert has suboptimal performance in the environment (or in causal language, in some SCM $M$) that is acting.
> > #### …
> > #### “It would be great if the authors can give some real cases that the performance of an expert is suboptimal even in the environment that is acting in it. The example in the introduction is not convincing enough as it is not clear why the expert acts based on just $X$ which results in poor performance.”
> > #### “... it is required to justify why the expert is not acting optimally even at least one SCM M.”
>
>
> This is a subtle point, thank you for the opportunity for us to clarify it. The condition $\nu^* \leq 0$ indicates that the expert’s policy might be suboptimal and the Causal IRL algorithm is able to find a policy that improves over the expert.
>
> Many real-world applications exist where the expert generating the demonstrations is possibly suboptimal, i.e., it’s an “expert” but not an “oracle,” and could therefore be improved. For instance, consider the development of autonomous vehicles, where the reinforcement signal is never fully known, and imitation learning methods are widely used. To collect demonstration data, it often requires a human demonstrator, possibly a driving expert, to drive the vehicle around the target environment (e.g., a city, a highway) and record the natural driving trajectories. No matter how clean their driving record is, the human demonstrator is naturally error-prone and cannot guarantee optimal driving behaviors at all times (Atchley et al., 2011). Indeed, in many complicated learning tasks, it might be difficult for the demonstrator to be consistently optimal, even for relatively veteran experts, e.g., playing complex video games or stock trading (Newell et al., 1972; Nikolaidis et al., 2017).
>
> Behavioral cloning methods mimic the demonstrator’s policy and it’s bound by the performance of this sub-optimal expert, and can never outperform it. On the other hand, by exploiting additional parametric knowledge about the latent reward signal, inverse RL methods are able to obtain an effective policy that consistently dominates the expert’s policy. This observation is not new, and was first proposed in (Syed and Schapire, 2008), who also formalized the use of performance gap $\nu^*$. (Zhang et al., 2020) studied the behavioral cloning from the combination of confounded observations and causal constraints about the environment. In this work, we take inspiration from inverse RL approaches and develop non-trivial imitation learning methods that are robust to both unobserved confounding and a sub-optimal expert.
>
> We have added the note above about $\nu^* \leq 0$ to our updated version of the manuscript (Appendix F, Q10); thanks for the suggestions.
>
> - Newell, Allen, and Herbert Alexander Simon. Human problem solving. Vol. 104. No. 9. Englewood Cliffs, NJ: Prentice-hall, 1972.
> - Nikolaidis, Stefanos, et al. "Game-theoretic modeling of human adaptation in human-robot collaboration." Proceedings of the 2017 ACM/IEEE international conference on human-robot interaction. 2017.
> - Atchley, Paul, Stephanie Atwood, and Aaron Boulton. "The choice to text and drive in younger drivers: Behavior may shape attitude." Accident Analysis & Prevention 43.1 (2011): 134-142.

---

> > ### Author Response · Authors · 2022-11-13
> > **Response to Reviewer U7KX [2/4]**
> >
> > > #### “The authors considered a ``sequential decision-making" setting; however, it seems that $Y$ is not indexed with time steps. How can the results in the paper be extended to the more general setting where we have a sequence of $Y_i$’s? In the FAQ, it is mentioned that $Y$ can be a set of variables but it might be the case that the cumulative reward is identifiable while each $Y_i$ is not identifiable from $P(O,Y)$.”
> >
> > We thank the reviewer for the question and would like to clarify the generalization to multiple reward signals further. As stated in Q8, Appendix D, “Theoretically, Thms. 1 and 2 hold when the reward is a set of variables, $\boldsymbol{Y}$. The reward function $r( \boldsymbol{x}^*, \boldsymbol{z}^*)$ is defined as the cumulative reward $\sum_{Y \in \boldsymbol{Y}} \mathbb{E}[Y \mid  \boldsymbol{x}^*, \boldsymbol{z}^*]$. For concreteness,  we demonstrate our algorithm in Experiment 4 to optimize the cumulative reward in a sequential decision-making problem with an infinite horizon.”
> >
> > Particularly, for a set of reward signals $\boldsymbol{Y}$, below we would like to point out that the cumulative reward is not identifiable as long as there exists one $Y_i$ that is not identifiable from $P(O,Y)$. If there is one single signal $Y_i \in \boldsymbol{Y}$ such that $E[Y_i |do(\pi)]$ is not identifiable, one could construct two SCMs $M_1$ and $M_2$ that coincide with all the reward signals in $\boldsymbol{Y}$ except $Y_i$. In this case, $M_1$ and $M_2$ define different evaluations of the cumulative reward $E[\sum_{Y_i \in \boldsymbol{Y}} Y_i |do(\pi)]$, while being compatible with the same causal diagram and the observational distribution. That means that the cumulative reward $E[\sum_{Y_i \in \boldsymbol{Y}} Y_i |do(\pi)]$ is not identifiable if there is one reward $Y_i \in \boldsymbol{Y}$ so that $E[Y_i|do(\pi)]$ is not identifiable.  For concreteness, consider a causal diagram with directed arrows $X \rightarrow Y_1$, $X \rightarrow Y_2$ and bidirected arrows $X \leftrightarrow Y_1$, $X \leftrightarrow Y_2$. In this diagram, the expected reward $E[Y_1 |do(\pi)]$ is not identifiable due to the bi-directed arrow $X \leftrightarrow Y_1$. We now show that the cumulative reward $E[Y_1 + Y_2 |do(\pi)]$ is also not identifiable. Consider two SCMs $M_1$, $M_2$ such that in $M_1$,
> > $$
> > X \gets U_1,\\
> > Y_1 \gets X \oplus U_1,\\
> > Y_2 \gets X \oplus U_2\\
> > $$
> >
> > And in $M_2$,
> > $$
> > X \gets U_1,\\
> > Y_1 \gets 0,\\
> > Y_2 \gets X \oplus U_2
> > $$
> >
> > $U_1, U_2$ are independent noise drawn uniformly from $\\{0, 1\\}$. It is verifiable that $M_1$ and $M_2$ generate the same observational distribution $P(X, Y_1, Y_2)$. For any atomic policy $\pi: X \gets x$, evaluating the cumulative reward in $M_1$ gives:
> > $$
> > E[Y_1 + Y_2 |do(x); M_1] = E[Y_1|do(x); M_1] + E[Y_2|do(x); M_1]\\
> > = E[U_1 \oplus x] + E[U_2 \oplus x] = 1
> > $$
> > The last step holds since $U_1, U_2$ are uniform binary variables. Evaluating the cumulative reward in $M_2$ gives
> > $$
> > E[Y_1 + Y_2 |do(x); M_2] = E[Y_1|do(x); M_2] + E[Y_2|do(x); M_2]\\
> > = 0 + E[U_2 \oplus x] = 0.5
> > $$
> > That is, the cumulative rewards $E[Y_1 + Y_2 |do(\pi)]$ is different in $M_1$ and $M_2$, therefore is not identifiable.
> >
> > Having said that, your point is well-taken and will be reflected in our discussion. We will  emphasize this point in an updated version of the manuscript.

---

> > > ### Author Response · Authors · 2022-11-13
> > > **Response to Reviewer U7KX [3/4]**
> > >
> > > ---
> > > > #### “Under what conditions, is $\nu^*$ positive? … The main ambiguity in the paper is for the case $\nu^*>0$.”
> > >
> > > When the performance gap $\nu^* > 0$, there exists at least one causal model such that the imitator can **never** achieve the expert’s performance.
> > >
> > > For instance, consider a bow graph consisting of a directed arrow $X \rightarrow Y$ and a bi-directed arrow $X \leftrightarrow Y$. We could construct a causal model $M$ compatible with this causal constraint such that no intervention $do(x)$ on action X could outperform the expert’s performance $E[Y]$. Let an exogenous variable $U$ be uniformly drawn over a binary domain $\\{0, 1\\}$. Let the expert’s policy be $X \gets U$ and the reward function $Y \gets X \oplus \neg U$. Evaluating the expert’s reward gives
> > > $$
> > > E[Y] = E[ X \oplus \neg U] = E[U \oplus \neg U] = 1
> > > $$
> > >
> > > On the other hand, for any intervention $do(x)$, the imitator’s reward is given by
> > > $$
> > > E[Y|do(x)] =  E[x \oplus \neg U] = 0.5
> > > $$
> > > The last step holds since $U$ is a uniform binary variable. In this environment, for any action, the imitator’s reward $E[Y|do(x)] = 0.5$ is far from the expert’s performance $E[Y] = 1$. This means that in the bow graph, the performance gap is at least $\nu^* > E[Y] - E[Y|do(x)] = 0.5$. Such settings are referred to as “non-imitable” in (Lemma 1, Zhang et al., 2020).
> > >
> > > We have added the note above about $\nu^* > 0$ to our updated version of the manuscript (Appendix F, Q11); thanks for the suggestions.
> > >
> > > ---
> > > > #### “What is the exact definition of effective actions and covariates in Theorem 1? What happens to other variables $\mathbf{X}$ and $\mathbf{Z}$ in Theorem 1?”
> > >
> > > The exact definition of effective actions and covariates is provided in Thm. 1, “effective actions $\boldsymbol{X}^* = \boldsymbol{X} \cap An(Y)\_{\mathcal{G}\_\mathcal{S}}$ and effective covariates $\boldsymbol{Z}^* = \bigcup_{X_i \in \boldsymbol{X}^*} \boldsymbol{Z}^*\_i$.” In other words, effective actions and covariates are observed variables that have causal relationships with the latent reward.
> > >
> > > It is sufficient to ignore actions and covariates outside $\boldsymbol{X}^*$ and $\boldsymbol{Z}^*$ since they have no causal relationship with the reward $Y$. For concreteness, consider a causal diagram with direct arrows $Z \rightarrow X_1 \rightarrow Y$, $Z \rightarrow Y$ and $Z \rightarrow X_2$. The effective action and covariate in this graph are $\{X_1\}$ and $\{Z\}$, respectively. Action $X_2$ could be ignored during the optimization process since there is no causal path from $X_2$ to $Y$, i.e., intervening on $X_2$ has no causal effect on outcome $Y$.

---

> > > > ### Author Response · Authors · 2022-11-13
> > > > **Response to Reviewer U7KX [4/4]**
> > > >
> > > >
> > > > ---
> > > > > #### “it is good to discuss the projection algorithm mentioned in the footnote on page 7 and also mention why it is required to perform such projections.”
> > > >
> > > > Indeed, and for clarity, we will also discuss it here. The projection algorithm marginalizes unobserved endogenous variables in a causal diagram and allows us to focus on the causal relationships among observed endogenous variables. It is particularly useful for causal identification since the identifiability of causal effects remains invariant across the projection operation. For instance, consider a causal diagram $G$ with directed arrows $X \rightarrow Z \rightarrow Y$ and a bi-directed arrow $X \leftrightarrow Y$; variable $Z$ is unobserved, so the observational distribution is given by $P(X, Y)$. In this case, one could simplify the diagram by projecting out node $Z$, i.e., $G’ = project(G, Z)$. The resulting graph $G’$ is given by $X \rightarrow Y$ and $X \leftrightarrow Y$. One could show that the interventional distribution $P(y|do(x))$ is not identifiable in the projected diagram $G’$, thus is not identifiable from $P(X, Y)$ in the original diagram $G$.
> > > >
> > > > In Sec 3, we could assume the causal diagram $G$ to be the outcome of the projection algorithm. Since the projection algorithm preserves causal relationships among observed endogenous variables, the identifiability results in Theorem 2 and Lemma 1 hold without loss of generality. We have added the pseudo-code for the projection algorithm and additional explanation in the updated manuscript (Appendix F, Q9). Thank you for the suggestion.
> > > >
> > > >
> > > > ---
> > > > > #### “It is not clear how causality can help to get better performance than the expert.”
> > > >
> > > > By utilizing causal relationships embedded in the environment, the learner is able to address the bias/distribution drift due to the presence of unobserved confounding in demonstration data. Once the adjustment formula is obtained (i.e., Thms. 1 and 2), IRL algorithms could produce a policy that substantially outperforms the expert by exploiting parametric knowledge about the reward, e.g., linearity (Syed and Schapire, 2008) or through reward augmentation (Li et al., 2017). In this work, one of our main contributions is to extend existing IRL approaches to more generalized settings where the expert’s demonstrations are imperfect, and the phenomenon of unobserved confounding exists (i.e., Markovianity cannot be assumed). To further ground our contributions, we summarize the current literature on imitation learning as follows, in order to highlight that our work fills a critical literature gap where unobserved confounders exist in the sequential imitating processes and the expert is sub-optimal:
> > > >
> > > > |     | Optimal Expert | Sub-optimal Expert|
> > > > | --- | ----------- |  ----------- |
> > > > | Unconfounded | Behavior Cloning | Inverse RL |
> > > > | Confounded | Causal BC | Causal IRL (**our work**)  |
> > > >
> > > > To sum up, behavior cloning (BC) is able to achieve satisfactory performance when the demonstration data is unconfounded, and the expert is (near) optimal. Inverse RL (IRL) can outperform the suboptimal expert by exploiting additional parametric knowledge about the reward signal. However, both BC and IRL are fragile to data drift due to the presence of unobserved confounders. Causal BC (Zhang et al., 2020; Kumor et al., 2021) produces a confounding-robust policy but is still limited by the performance of a suboptimal expert. Finally, this paper empowers IRL methods with the causal inference theory so that the IRL imitator could produce a policy robust to both unobserved confounding and the suboptimal expert.

---

> > > > ### Comment · Reviewer_U7KX · 2022-11-14
> > > > **About the review response**
> > > >
> > > > Thank you for addressing the comments and also revising the paper. My comments were fairly addressed. Regarding the example for $\nu^*>0$, it seems that $E[Y]-R[Y|do(x)]=-0.5$. Am I right? BTW, it would be great if the authors could change the example in the introduction such that the expert is using both $X$ and $Z$ but it has poor performance with respect to causal IRL.

---

> > > > > ### Author Response · Authors · 2022-11-16
> > > > > **Thanks for your suggestions**
> > > > >
> > > > > We appreciate the reviewer’s response and helpful suggestions; thank you.
> > > > >
> > > > > ---
> > > > >
> > > > > > #### “Regarding the example for $\nu^*>0$, it seems that $E[Y]-R[Y \mid do(x)]=-0.5$, Am I right?”
> > > > >
> > > > >
> > > > > Good catch, indeed, there is a small typo in the previous version and we apologize for that. We updated the corresponding “bow graph” example in “Response to Reviewer U7KX [3/4]”. The reward function should be $Y \gets X \oplus \neg U$. The expert’s policy remains the same as $X \gets U$. In this situation,
> > > > >
> > > > > The expert’s reward:
> > > > > $$
> > > > > E[Y] = E[ X \oplus \neg U] = E[U \oplus \neg U] = 1
> > > > > $$
> > > > >
> > > > > The imitator’s reward:
> > > > > $$
> > > > > E[Y|do(x)]
> > > > > $$
> > > > >
> > > > > $$
> > > > > =  E[x \oplus \neg U]
> > > > > $$
> > > > >
> > > > > $$
> > > > > = 0.5\*E[x] + 0.5\*E[\\neg x] = 0.5\*E[x] + 0.5\*(1-E[x]) = 0.5
> > > > > $$
> > > > >
> > > > > Therefore, the performance gap is at least $\nu^* > E[Y] - E[Y|do(x)] = 0.5$. This means that for any action, the imitator’s reward $E[Y|do(x)] = 0.5$ is far from the expert’s performance $E[Y] = 1$, i.e., the expert’s performance is “non-imitable”.
> > > > >
> > > > >
> > > > > ---
> > > > > > #### “It would be great if the authors could change the example in the introduction such that the expert is using both $X$ and $Z$ but it has poor performance with respect to causal IRL.”
> > > > >
> > > > >
> > > > > Yes, we agree that “it would be great if the authors could change the example in the introduction such that the expert is using both $X$ and $Z$”. We have updated the first example in our updated manuscript. Specifically, the “human expert generates demonstrations following a behavior policy such that $P(X = 1 \mid Z = 0) = 0.6$ and $P(X = 0 \mid Z = 1) = 0.4$. Evaluating the expert's performance gives $\mathbb{E}[Y] = P(X= 1, Z = 0) + P(X = 0, Z = 1) = 0.5$” (also discussed on page 2, Line 1). In this case, the expert’s performance remains the same $\mathbb{E}[Y] = 0.5$, and IRL is able to produce a policy that outperforms the sub-optimal expert.

---

### Author Response · Authors · 2022-11-26
**Summary of Revisions**

We thank all the reviewers for their willingness to engage and the valuable and constructive feedback provided throughout the process. We have revised the paper and have addressed all the concerns raised during the reviewing and discussion stages. As always, we welcome your suggestions and comments and would love to improve the manuscript further. Changes in the updated manuscript are summarized below:


1. **Updated motivating examples**: we updated the first example in Sec. 1 to better explain our focal message that causality is needed for practical IRL.
2. **Highlighted definitions**: We highlighted “atomic intervention” and “policy intervention” on Page 3.
3. **Highlighted goal**: we added a formal definition of imitating policy in Definition 1 (Page 4), which highlights the goal of the paper.
4. **Highlighted input and output**: we highlighted the input and output of the agent (the imitator) on Page 4, Lines 2-4; we also explicitly listed the input, output, and objective of our interested problem in Q13, Appendix F.
5. **Clear references/pointers to algorithms**: we added a clearer reference to Causal MWAL (Alg. 3) and Causal GAIL (Alg. 4) on Page 6.
6. **Structure/organization of algorithms**: we added more details in Algs. 1-4 to explicitly exhibit the input and the output for each algorithm.
7. **Projection algorithm**: we added the pseudo-code for _the projection algorithm_ and additional explanations in Appendix F, Q9.
8. **Discussions about $\\nu^\*$**: we provided more discussions about $\nu^*$, specifically, $\nu^\* \leq 0$ (Appendix F, Q10) and $\nu^\* > 0$ (Appendix F, Q11).
9. **Important role/contribution of causality in IRL**: we added more explanations about “how causality helps inverse RL” in Appendix F, Q12, and highlighted our contributions further.
10. **Notations table**: we added Table 2 in Appendix F, summarizing important notations used in this paper.

---
We would like to note that one of the contributions of this paper is to“empower existing IRL methods with the causal inference theory so that the IRL imitator could produce a policy robust to both unobserved confounding and the suboptimal expert”. We believe that a more principled backing for IRL through the lenses of causality is a critical component of RL foundations, which has been validated through extensive and systematic experimentation.

---

### Decision · Program_Chairs · 2023-01-20

**Decision:**

Accept: poster

**Justification For Why Not Higher Score:**

The theory seems to be a direct extension of existing work in the behavioral cloning setting. As such, it is unclear if the contributions are significantly novel. The experiments are also on somewhat simple benchmarks, which also makes scalability to more realistic scenarios unclear.

**Justification For Why Not Lower Score:**

There are no major outstanding concerns among the reviewers and the work seems to address a relevant problem.

**Metareview: Summary, Strengths And Weaknesses:**

This paper studies inverse RL in the presence of unobserved confounding. The primary use case in this context is imitation learning. Previously, this analysis has been done for direct imitation via behavioral cloning and the author's contribution here extends it to indirect/inverse RL approaches.

The paper has 2 key contributions. First, the paper analyzes structural conditions on the causal model under which learning the expert policy is possible in the presence of unobserved confounding. Second, the authors further exploit knowledge of the graphical structure to extend IRL algorithms such as GAIL to the confounded setting. Some empirical evaluations on real and synthetic settings are also provided. The reviewers generally held consensus on the merit of the work in terms of clear and convincing theory. There were some questions regarding the assumptions, empirical setups,  and self-contained background of the work, all of which were adequately addressed by the authors. I recommend acceptance.


**Note From Pc:**

if the above contains the word "oral" or "spotlight" please see: "oral" presentation means -> notable-top-5% and "spotlight" means -> notable-top-25%. As stated in our emails, we are disassociating presentation type from AC recommendations